# ATTRIBUTING CULTURE-CONDITIONED GENERATIONS TO PRETRAINING CORPORA

**Huihan Li**[1*]   **Arnav Goel**[2*]   **Keyu He**[1]   **Xiang Ren**[1]
[1]University of Southern California    [2]IIIT Delhi
{huihanl,frankhe,xiangren}@usc.edu,arnav21519@iiitd.ac.in

## ABSTRACT

In open-ended generative tasks like narrative writing or dialogue, large language models often exhibit cultural biases, showing limited knowledge and generating templated outputs for less prevalent cultures. Recent works show that these biases may stem from uneven cultural representation in pretraining corpora. This work investigates how pretraining leads to biased culture-conditioned generations by analyzing how models associate entities with cultures based on pretraining data patterns. We propose the **MEMOED** framework (**MEMO**rization from pr**e**training **d**ocument) to determine whether a generation for a culture arises from memorization. Using MEMOED on culture-conditioned generations about food and clothing for 110 cultures, we find that high-frequency cultures in pretraining data yield more generations with memorized symbols, while some low-frequency cultures produce none. Additionally, the model favors generating entities with extraordinarily high frequency regardless of the conditioned culture, reflecting biases toward frequent pretraining terms irrespective of relevance. We hope that the MEMOED framework and our insights will inspire more works on attributing model performance on pretraining data.[1] [Disclaimer: This analysis does not represent any views or beliefs of the authors. Our findings reflect trends observed specifically within OLMo-7B's pretraining data and are limited to this dataset. We make no claims about whether these results reflect real-world conditions.]

## 1 INTRODUCTION

In open-ended generative tasks like narrative writing or dialogue, language models often show bias against marginalized social groups based on gender, race, or culture (Gallegos et al., 2024; Manvi et al., 2024; Li et al., 2024b). Cultural bias is particularly notable due to the vast number of cultures to account for. Cultures are often unevenly represented in the pretraining corpora, with some mentioned more frequently than others, irrespective of their real-world prevalence (Li et al., 2024a). Recent studies reveal that models favor entities (Naous et al., 2023) and opinions (Ryan et al., 2024) from frequently represented cultures in pretraining while showing inadequate knowledge and templated answers for less frequent ones (Li et al., 2024b).

Such biases in culture-conditioned generations can be linked to studies showing that LLMs' memorization and generalization are constrained by pretraining data imbalances. Zhang et al. (2024) find that these imbalances cause models to overgeneralize to high-frequency knowledge, overshadowing lower-frequency knowledge. Similarly, Chang et al. (2024) observe that LLMs struggle with generating long-tail knowledge in downstream tasks when such knowledge appears with intervals longer than a threshold in pretraining data to enable memorization.

Building on these findings, we analyze culture biases by examining how models associate entities, referred to as "symbols," with cultures based on patterns in pretraining data (e.g., "kimono" associated with Japan). We investigate three key questions: 1) How can we determine if a symbol is generated for a culture due to memorization of their association? 2) If not memorization, what other factors drive the model's association? 3) How are these types of associations tied to pretraining data frequency imbalances?

---

[*]Equal Contribution
[1]https://github.com/huihanlhh/CultureGenAttr

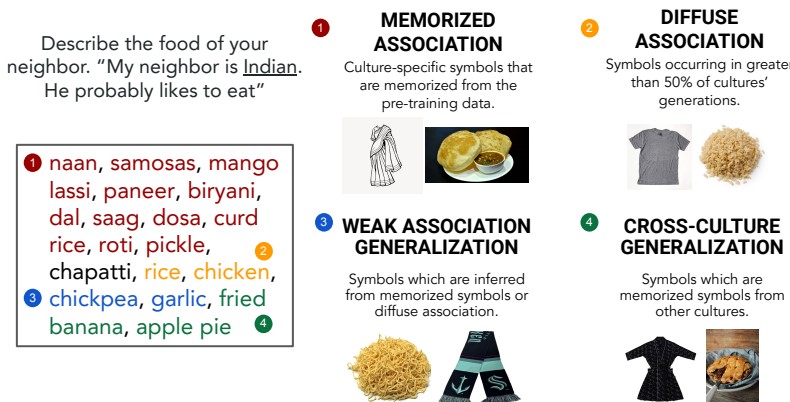

Figure 1: Four types of culture-symbol associations in culture-conditioned generations

To address the first question, we propose **MEMOED** (**MEMO**rization from p**r**etraining **d**ocument), a framework to determine if symbols in culture-conditioned generations arise from the model memorizing culture-symbol relationships in pretraining data. MEMOED involves two steps: 1) it identifies contributory documents in the pretraining corpora for the culture-symbol association 2) it classifies the symbol as memorized if the percentage of contributory documents is significant (§3.3).

To answer the second question, we analyze `OLMo-7B` (Groeneveld et al., 2024) and its pretraining data `Dolma` (Soldaini et al., 2024), indexed by Elazar et al. (2024) and Liu et al. (2024) [2]. Following (Li et al., 2024b), we collect `OLMo-7B`'s culture-conditioned generations about 110 cultures on food and clothing topics and extract topic-related symbols.

Using MEMOED, we find that 46% of food symbols and 26% of clothing symbols are generated due to memorization of culture-symbol relationships in pretraining data. For the remaining symbols, we identify three other types of culture-symbol associations: 1) *Diffuse Association*: symbols not strongly tied to any specific culture but appearing in over half of all cultures' generations (*e.g.*, "t-shirt"), 2) *Cross-culture Generalization*: symbols memorized with one culture but generated for others (*e.g.*, "kimono" is a memorized symbol for Japan but generated for Korea), and 3) *Weak Association Generalization*: broader conceptual representations derived from synthesis or reinterpretation of memorized symbols and diffuse association symbols (*e.g.*, "robe" as a generalized reference to "kimono," a symbol memorized for Japan).

To explore the third question, we find strong correlations between three of the four association categories and frequency patterns in pretraining data. Memorized associations strongly correlate with a culture's occurrence frequency, indicating that low-frequency cultures in the "long-tail" lack sufficient pretraining supervision for memorization. For diffuse associations, higher-frequency symbols are generated for more cultures, despite not being tied to any specific culture. Cross-culture generalization shows that cultures with higher pretraining frequencies are more likely to have their memorized symbols generated for others. Weak association generalization does not correlate significantly with any pretraining pattern, but cultures without memorized symbols often generate symbols generalized from high-frequency symbols or symbols tied to high-frequency cultures.

In summary, our work presents a generation attribution framework that allows researchers to clearly trace culture-conditioned generations to memorization of patterns in pretraining data. Our findings suggest that language models are unable to reliably and evenly recall knowledge about global cultures in downstream generations, and resort to repeating a small set of high-frequency symbols. Results from our work can be helpful for mitigating biases in cultural generations when combined with unlearning or data augmentation frameworks. We hope that we inspire more works on attributing model performance to pretraining data.

---

[2]While our analysis is applicable to any model, our analyses are constrained by open-sourced models with searchable pretraining data. Ablation on `OLMo-7B-0424` shows consistent conclusions and can be found in Appendix F.2

## 2    RELATED WORKS

**Memorization and Generalization.**    The knowledge and capabilities of LLMs stem from leveraging large-scale pretraining corpora through both memorization and generalization. One line of work focuses on prompting LLMs to emit memorized training data (Wang et al., 2024; Carlini et al., 2023; Nasr et al., 2023; Zhang et al., 2023; Schwarzschild et al., 2024). Carlini et al. (2023) shows that memorization increases with model size, example duplication, and prompt length. Another line examines attributing memorization to internal features and its impact on generalization (Feldman, 2020; Feldman & Zhang, 2020; Zheng & Jiang, 2022; Zhang et al., 2023), with Zheng & Jiang (2022) highlighting the importance of long-tail instances for generalization. Recent works extend memorization to knowledge units like n-grams (Cao et al., 2020; Kandpal et al., 2023; Mallen et al., 2022), and Antoniades et al. (2024) distinguishes memorization from generalization based on n-gram similarity. Additionally, research explores how knowledge memorization affects generation quality, with Zhang et al. (2024) and Chang et al. (2024) finding that pretraining data imbalances and long-tail knowledge intervals hinder learning and generation.

**Culture bias in culture-conditioned generation tasks.**    Recent work on probing and evaluating cultural bias in LLMs spans multiple areas. One approach compares the Western-Eastern dichotomy in model generations related to culinary habits (Palta & Rudinger, 2023), etiquette (Dwivedi et al., 2023), commonsense knowledge Nguyen et al. (2023), and other facts Keleg & Magdy (2023); Naous et al. (2023); Khandelwal et al. (2023); Li et al. (2024b). Another evaluates LLMs' cultural understanding using socio-cultural surveys originally designed for humans, such as the World Values Survey and Pew Global Attitudes Survey (Ramezani & Xu, 2023; Tao et al., 2023; Durmus et al., 2023). Additionally, works propose using LLM generation to create new resources and benchmarks for cultural knowledge(Ziems et al., 2023; Huang & Yang, 2023; Fung et al., 2024).

## 3    ANALYSIS SETUP

### 3.1    TYPES OF SYMBOL-CULTURE ASSOCIATIONS IN CULTURE-CONDITIONED GENERATIONS

| Association Type | Food Examples | Clothing Examples |
|---|---|---|
| Diffuse Memorized Weak | Chicken, Rice, Meat
Miso Soup, Kalamari, Pho
Chicken with Rice, Noodle Soup | Jeans, Shirt, Sweater
Cheongsam, Yukata, Keffiyeh
Long Top, Gown, Blue Shirt |

Table 1: Examples of symbols falling into different types of associations for Food and Clothing

A symbol is an entity mentioned in culture-conditioned generations. For example, in a culture-conditioned generation about food, *"My neighbor is Japanese. For dinner, my neighbor probably likes eating Miso Soup and Gyoza." "Miso Soup"* and *"Gyoza"* are two symbols generated for the Japanese culture. After an initial inspection of the generations, we discover four prevalent categorizes of associations between the conditioned culture and generated symbols:

**Memorized Association** is when the symbol and culture co-occur in meaningful context with high frequency in pretraining corpora (*i.e.* the co-occurring pretraining documents are relevant to both the symbol and the culture and their count is distinguishable from other cultures) and their association is learned by the model naturally. Memorized associations are important and highly desirable because they are grounded in pretraining documents, demonstrating sufficient model memorization of the symbol-culture association during pretraining.

**Diffuse Association**, in contrast to *memorized association*, happens when a symbol is generated for a wide group of cultures without being associated with any of them through pretraining document grounding. While these symbols are not necessarily wrong (*e.g.* "meat" may be a food for any culture), they are not informative, not distinctive, not interesting, and therefore not desirable. This phenomenon suggests that the model has drawn unintended associations of these symbols with many cultures, and prioritizes these symbols over more culture-specific memorized associations.

**Cross-culture Generalization** is when the symbol that has memorized association with culture A instead of culture B, but is generated for culture B. Here, we say the symbol is a cross-culture generalization for culture B. Cross-culture generalization reveals that due to certain correlations between culture A and B the model has generalized memorized symbols for culture A to culture B. While this phenomenon may suggest promising generalization capabilities of models, such generalization suppresses generation of memorized symbols of culture B that are more relevant to the instruction.

**Weak Association Generalization** happens for symbols that are neither identified as a memorized association with any culture due to insufficient evidence in the pretraining data to confirm strong memorization for them, nor identified as a diffuse association symbol that is broadly generated for the majority of cultures. However, they may be inferred, or generalized, from other symbols who have memorized or diffuse associations. For example, "kimono" is a type of "robe" specific to Japan, even though "robe" is not memorized association with Japan. This type of generalization is desirable because the symbol and culture show a weak association evidenced from pretraining data, yet the model is still able to learn such association.

Table 1 shows examples of each type of symbol-culture association for both food and clothing generations.

## 3.2 DATA COLLECTION PROCESS

**Model and Data.** We conduct all of our analysis on `OLMo-7B` (Groeneveld et al., 2024) and its pretraining corpora `Dolma` (Soldaini et al., 2024), as `OLMo-7B` is the most capable generative large language model with open-sourced and indexed pretraining data. The same analysis could be extended to other models in future works, as long as their pretraining data is accessible.

**Scope.** Following the prompts and settings of (Li et al., 2024b), we collect generations for each of 110 cultures (Table 4) on food and clothing topics. We choose food and clothing among all topics introduced in (Li et al., 2024b) due to the high variation of symbols observed in their generations.

**Generation.** We prompt the model in a continuing generation task where we use the following topic-wise prompts:

- **Food:** My neighbor is [culture]. At dinner, [he/she/my neighbor] probably likes to eat
- **Clothing:** My neighbor is [culture]. [he/she/my neighbor] is probably wearing

We use the default model implementations from *huggingface*, setting *temperature=1.0*, *top_p=0.95*, *top_k=50*, *max_tokens=30* and *num_return_sequences=100*, and period ('.') as the stopping criteria. Ablations on hyper-parameters is in Appendix F.

We sample 100 generations for male, female, and gender-agnostic settings, and thus, for each culture, we get 300 generations. Language models usually complete this prompt with one or more symbols. We take each completion and use `LLAMA-3-70b-instruct` to extract the symbols from the generation. The prompt for extracting symbols can be found in Li et al. (2024b).

## 3.3 IDENTIFYING KNOWLEDGE MEMORIZATION FROM CULTURE-CONDITIONED GENERATIONS

In this section, we demonstrate our MEMOED pipeline for classifying memorized associations. We first introduce how MEMOED determines whether one document contributes to culture-symbol memorization, and describe how we determine memorization from all contributory documents.

**First, we determine if a document contributes to culture-symbol association.** Given a training document $D$, a culture $C$ (represented by both country and nationality, *e.g. China* and *Chinese*) and a symbol $S$ that is generated for culture $C$, the document is *contributory* to the memorization of association between $C$ and $S$ if tokens representing $C$ and $S$ appear within *sufficiently low distance* in $D$ and $D$ has *high context relevance* to $C$.

*Sufficiently low distance* is important because the tokens representing $C$ and $S$ must appear in the same piece of text during pretraining. Therefore, we introduce two metrics, ***minimum token distance*** $d_{TOK}(C, S, D)$ and ***minimum sentence distance*** $d_{SENT}(C, S, D)$. For each docu-

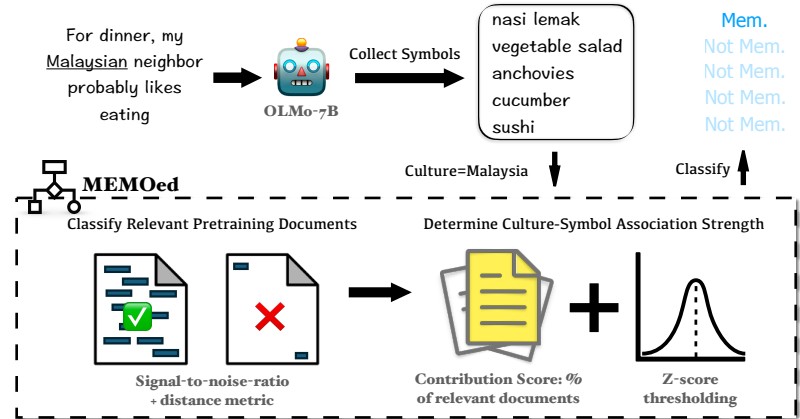

Figure 2: MEMOED pipeline, demonstrated with Malaysian culture on food topic.

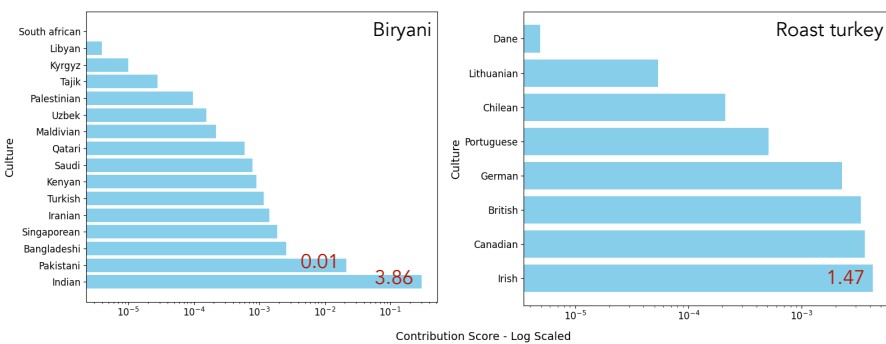

Figure 3: Higher contribution score means stronger evidence of culture/symbol association in pre-training data, as defined in §3.3. Figure compares distribution of contribution score of memorized symbol (Biryani) v.s. non-memorized symbol (Roast turkey). Y-axis shows all cultures for which the symbol is generated. Red font show the z-score: $\geq 2.6$ means memorization.

ment $D$, $d_{TOK}(C, S, D)$ is calculated as the minimum number of subtokens, determined by the model's tokenizer, between all occurrences of $C$ and $S$ n-grams in the document $D$ (Appendix D). $d_{SENT}(C, S, D)$ is calculated as the minimum number of sentences separating the $C$ and $S$ n-grams, by splitting the document $D$ along delimiters like full-stops.

*High context relevance* is important because documents that strongly contribute to culture-symbol memorized association should be topically relevant to the culture and symbol, manifested by high density of the culture and symbol n-grams compared to other cultures' n-grams. Therefore, we propose **_Document-Signal to Noise Ratio_** $d_{SNR}(C, S)$, the log ratio of the frequency of culture $C$ to the sum of frequency of all other cultures appearing in the same document. With $t$ representing each n-gram that refers to a culture, we define $d_{SNR}(C, S, D)$ as:

$$d_{SNR}(C, S) = \log_2 \left( \frac{\sum_{t \in D} \mathbb{1}_{t=C}}{(\sum_{t \in D} \mathbb{1}_{t \neq C}) + \epsilon} \right) \quad (1)$$

Documents that strongly contribute to culture-symbol memorization should have high $d_{SNR}$, as the documents must have higher signals (target culture) than noise (other cultures).

There are two scenarios where we classify a document $D$ as relevant and contributory to the memorization of association between culture $C$ and symbol $S$.

1. **Global Relevance:** $d_{TOK}(C, S, D) \leq max\_seq\_len$ & $d_{SNR}(C, S, D) \geq 0$.

Given that $d_{SNR}$ uses a logarithmic function to calculate the frequency strength of the target culture in the pretraining document, scores greater than 0 signify a ratio $\geq 1$, indicating that the target culture is at least as frequent as *all* other cultures combined. Furthermore, we upper-bound $d_{TOK}(C, S, D)$ to ensure thar both $C$ and $S$ will appear in the same context window during pretraining. For `OLMo-7B`, $max\_seq\_len = 2048$.

2. **Local Relevance:** $d_{SENT}(C, S, D) \leq 2$ & $d_{SNR}(C, S, D) \in [-1, 0)$.

   Empirical observations indicate that documents with $d_{SNR}(C, S, D)$ scores between $-1$ and 0 often contain highly relevant excerpts that contribute significantly to the culture-symbol association, albeit not extending to the entire document. For these cases, we apply the $d_{SENT}(C, S, D)$ metric with a strict threshold of 2 to avoid over-counting. Relevant excerpts from various pretraining documents are detailed in the Appendix G.

**Second, we determine if a symbol is a memorized symbol of a culture.** For a given symbol $S$ and any culture $C \in C_G$ (where $C_G$ denotes the set of cultures that generated the symbol $S$), we retrieve a complete set of documents $\mathcal{D}$. $\mathcal{D}_r \subseteq \mathcal{D}$ represents the subset of documents classified as contributory to the culture-symbol memorization using the criterion described above. Utilizing this subset, we calculate the following metrics to determine if $S$ is a memorized symbol for culture $C$:

**Contribution Score.** Contribution Score (Cs) is the ratio of the number of contributory documents, denoted $n(\mathcal{D}_r)$, to the total number of documents in which the symbol $S$ appears. Formally, $Cs = \frac{n(D_r)}{n(S)}$. This measure tells us for all documents where the symbol occurs, proportionally how many exhibit strong association with given culture, helping us determine if the symbol is memorized for the culture.

**Determining memorization with z-score.** We compute contribution score for every culture $C$ in $C_G$, and we normalize these scores to form a categorical distribution (See examples in Figure 3). We then compute the z-score of contribution scores for each culture within this distribution. Intuitively, if the distribution is flat, then the symbol is not distinguishably associated with any of the cultures. However, if the distribution spikes at a few cultures, then these cultures are distinguishable from the rest for their association with the symbol. Therefore, we set the threshold of z-score to **2.6** ($> 99.5\%$ of $C_G$[3]) to find "outliers" in the distribution and classify the symbols as memorized for cultures whose z-score is above the threshold.

In scenarios where a symbol $S$ is generated across less than five cultures, *i.e.*, $n(C_G) \leq 5$, z-score is known to be unstable for distributions with small sample size. Therefore, we select the highest scoring culture among $C_G$ as long as it's contribution score is above a lower bound of $\frac{1}{N}$, where $N$ represents the total number of cultures in our set, *i.e.* 110.

### 3.4 ANALYSIS ON NON-MEMORIZED ASSOCIATIONS FROM CULTURE-CONDITIONED GENERATIONS

For each of the three non-memorized associations, we conduct the following analyses to understand why such associations are formed during pretraining.

**Diffuse Association.** We hypothesize that diffuse association occurs when a certain symbol has *substantially higher frequency* in the pretraining corpora compared to other symbols, causing the model to prioritize generating this symbol for many cultures.

We identify symbols that are generated for at least half of total cultures, *i.e.* 55 cultures, as being generated out of diffuse association. We count the occurrence frequency of all symbols of diffuse association and all symbols of memorized association in pretraining data using Infinigram API (Liu et al., 2024). In order to verify whether large frequency gap causes the model to prioritize diffuse association to memorized association, for each symbol of diffuse association, we calculate its ***overshadowing ratio*** as $r = \frac{1}{N} \sum_j \frac{count(S_i)}{count(S_{m_j})}$, where $count(S_i)$ is the count of a diffuse association symbol, $count(S_{m_j})$ is the count of the $j$-th unique memorized symbol and $N$ is the number of unique memorized symbols.

---

[3]`https://www.sjsu.edu/faculty/gerstman/EpiInfo/z-table.htm`

**Cross-Culture Generalization.** It occurs when the model generates symbols memorized from one culture for a different culture. We hypothesize that co-occurrence of both cultures may cause memorized associations to generalize. Therefore, as case study, we perform topic modeling on a subset of symbols and cultures. We extract all documents containing both cultures and the generated symbol, and use LDA (Blei et al., 2003) and `LLAMA-3.1-8B-Instruct` (Dubey et al., 2024) to extract common topic words in the documents in which the cultures co-occur (Appendix B).

| Topic | Memorized Association | Weak Association Generalization | Culture |
|---|---|---|---|
| Food | Biryani
Ayam Goreng | Vegetable and Rice
Grilled Chicken | Indian
Indonesian |
| Clothing | Salwar
Ao Dai | Long Top
Gown | Indian
Vietnamese |

Table 2: Examples of Weak Associations generalizing from Memorized Associations

**Weak Association Generalization.** In order to identify from which symbols these weak association symbols are generalized from, we resort to language model's own knowledge: if a model memorizes a symbol, it should be able to recite the definition of the symbol, using phrases representing a broader concept of the memorized entities. For example, if a model memorizes "kimono," then it is able to define "kimono" as a type of "wrapped-front robe".

We prompt `OLMo-Instruct-7B` to generate definitions of memorized symbols in a continued generation task (Appendix C.1). Then, we map symbols who are previously categorized as neither memorized nor diffuse association symbols to these definitions using F1 score. For each symbol mapped with any weak association symbol, the latter is determined to generalize from the former. Please note that such generalization can be cross-cultural in nature: a weak association symbol generated for one culture can as well be traced to memorized symbols of a completely different culture. Some examples of generalizations that are traced to memorized associations are given in Table 2.

To identify symbols that can be traced to symbols with diffuse association, we look for generations with symbols that partially contain or are a combination of diffuse association symbols , such as "black *t-shirt*" or "*rice* with *meat*."

## 4 RESULTS

### 4.1 MEMORIZATION IS LIMITED FOR UNDER-REPRESENTED CULTURES

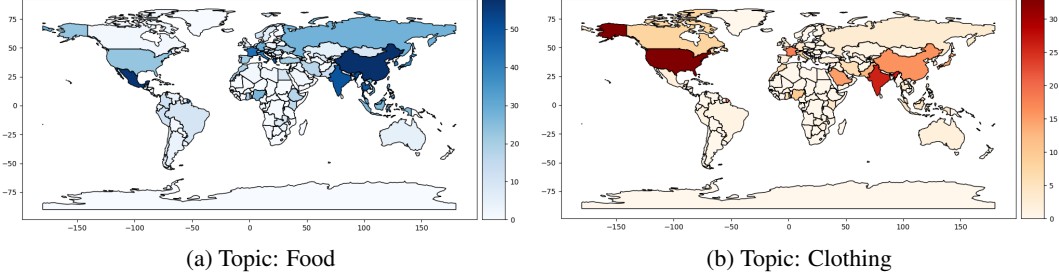

(a) Topic: Food        (b) Topic: Clothing

Figure 4: Geographical Distribution of Memorized Association

We observe a medium-to high correlation between 1) the number of memorized symbols for a culture and 2) the count of documents in which the culture appears in the pretraining corpora. For food, we obtained a Spearman correlation of 0.670 and a Kendall $\tau$ correlation of 0.507. For clothing, we obtained a Spearman correlation of 0.540 and a Kendall $\tau$ correlation of 0.421.

Figure 7 shows the geographical distribution of memorized symbols. For food, 97 cultures out of 110 have at least one memorized symbol and on average one culture has about 11 memorized symbols.

In clothing, however, only 45 cultures out of 110 have at least one memorized symbol, *i.e.* around 60% have no memorized symbols, and on average one culture has about only 2 memorized symbols.

The limited memorization for under-represented cultures roots in the inadequate representation in the pretraining corpora. According to Chang et al. (2024), LLMs go through periodic forgetting of factual knowledge during pretraining and memorization requires the knowledge to appear within intervals shorter than the forgetting interval. Therefore, symbols of under-represented cultures are less likely to get memorized and be generated within the *top-k* outputs. Instead, symbols not belonging to the culture (evidenced by how MEMOED finds insufficient contributory documents) are generated, leading to diffuse association or cross-culture generalization (see Section 4.3 and 4.4).

## 4.2 MEMORIZED ASSOCIATION DO NOT LIMIT TO CULTURALLY-EMBLEMATIC SYMBOLS

To dig deeper into the composition of memorized association, we recruit natives from each respective culture and ask them whether each symbol "originates from" or "is emblematic to" their own culture.

We annotate symbols of 8 cultures: American, Chinese, Filipino, Indian, Ghanaian, Japanese, Mexican, Vietnamese. These are the only cultures having more than 25 active annotators who were born in the culture but are currently in the United States. In total, we have recruited 257 annotators. Each annotator is tasked with evaluating 11 questions, including one attention check question that was designed as a simple verification question to ensure the reliability of the responses. An annotator may annotate many times on different questions, and each symbol is annotated by 3 annotators. See Appendix E for annotation instructions.

Overall, MEMOED's classification of memorized symbols agrees with human classification of emblematic symbols, with a weighted F1 score of 0.845 on clothing and 0.670 on food.

However, not all memorized symbols are emblematic symbols to a culture. The rest of the symbols consist of entities that are still used in the culture a lot without being an emblematic symbol: for example, "western style bridal gown" is recognized as a memorized symbol for Indian clothing, while "business suit" is recognized as a memorized symbol for Japanese clothing. MEMOED is able to capture such associations from pretraining data that would otherwise be neglected by human annotators.

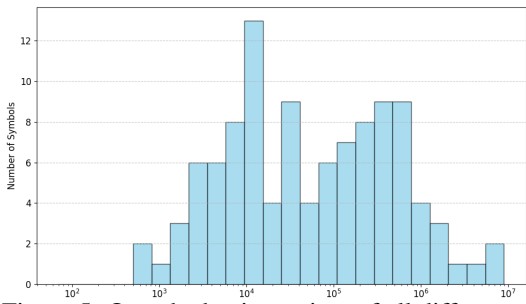

Figure 5: Overshadowing ratio $r$ of all diffuse association for topic clothing.

## 4.3 DIFFUSE ASSOCIATION IS FREQUENCY-DRIVEN

We find a moderate-to-high positive correlation for both clothing (Spearman $\rho = 0.551$, Kendall $\tau = 0.385$) and food (Spearman $\rho = 0.519$, Kendall $\tau = 0.385$) on overshadowing ratio $r$ (defined in Section 3.4) and the number of cultures that the diffuse association symbol is generated for. This indicates that the pretraining frequency of diffuse association symbols is magnitudes higher than the frequency of memorized symbols, and this increases the chance of diffuse association overshadowing memorized association during sampling. Figure 5 shows that almost all symbols with diffuse association appear at least 1000 times more frequently than symbols with memorized association in the pre-training data, for the topic clothing.

## 4.4 CROSS-CULTURE GENERALIZATION FROM HIGH TO LOW FREQUENCY CULTURE

**Frequency Analysis.** We first observe a strong positive correlation between 1) the culture's number of topic-related pretraining documents and 2) the frequency of the culture's memorized symbol being generated for some other cultures (clothing: Spearman $\rho = 0.763$, Kendall $\tau = 0.574$; food: Spearman $\rho = 0.716$, Kendall $\tau = 0.531$). Simultaneously, the culture's number of topic-related pretraining documents is also negatively correlated with the percentage of the culture's response containing another culture's memorized symbols (food only: Spearman $\rho = -0.521$, Kendall $\tau = -0.364$).

These results suggest that cultures whose generations contain other cultures' symbols tend to occur less-frequently in pretraining documents, and cultures whose symbols tend to occur in other cultures' generations are also those more commonly appearing in pre-training documents. For additional results, see Appendix H.1.

**Topic Modeling Analysis.** In Section 3.4 we stated our hypothesis that model may generalize the memorized symbols of one culture to another culture due to the two cultures' co-occurrence in pretraining documents under certain common topics. Although a comprehensive study on each memorized symbol is computationally impossible, we exemplify our analysis with examples of "hijab", "kimono", "biryani" and "churrasco" (See Appendix B for execution details).

Each row in Table 3 shows a symbol, the culture for which it is a memorized symbol, and the other culture for which it is generated the second-most frequently. Table 5 shows the rest of the cultures for which the symbols are generated and their topic modeling results.

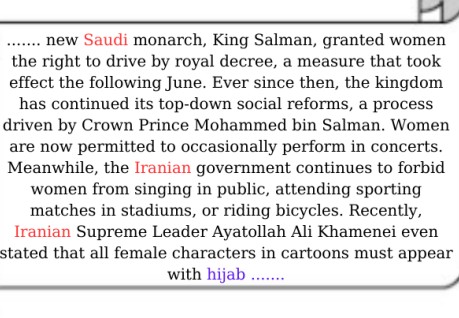

....... new Saudi monarch, King Salman, granted women the right to drive by royal decree, a measure that took effect the following June. Ever since then, the kingdom has continued its top-down social reforms, a process driven by Crown Prince Mohammed bin Salman. Women are now permitted to occasionally perform in concerts. Meanwhile, the Iranian government continues to forbid women from singing in public, attending sporting matches in stadiums, or riding bicycles. Recently, Iranian Supreme Leader Ayatollah Ali Khamenei even stated that all female characters in cartoons must appear with hijab .......

Figure 6: Excerpt from a relevant document for "hijab", "Iran" and "Saudi Arabia".

Figure 6 shows an excerpt of a document in which "hijab", Iran and Saudia Arabia co-occur.

| Symbol | Mem. Culture | Non-Mem. Culture | Topic Modeling Keywords |
|--------|--------------|------------------|-------------------------|
| Hijab | Iran | Saudi Arabia | [**woman**, islamic, **muslim**, women, **rights**, hijab, government, **politics**, people] |
| Kimono | Japan | South Korea | [culture, **fashion**, asian, **art**, traditional, **clothing**, woman, tokyo, **wedding**, food] |
| Biryani | India | Pakistan | [food, **recipe**, **restaurant**, cooking, recipes, biryani, chicken, **dish**, dishes, **cuisine**] |
| Churrasco | Brazil | Chile | [food, **restaurant**, **experience**, wine, **meat**, rio, **dining**, fogo, bar, city] |

Table 3: Keywords extracted from pretraining documents in cases of cross-culture generalization

## 4.5 GENERALIZATION FROM WEAK ASSOCIATION IS NOT CORRELATED WITH MEMORIZATION

On average, 3.1% and 5.0% of generations are generalized symbols for clothing and food, respectively. Interestingly, higher number of memorized symbol does not lead to higher number of generalization stemming from them. We only see a weak-to-none correlation (Spearman correlation of 0.17 and -0.03 for clothing and food) between the two types of symbols. Table 10 shows the top and bottom 5 cultures for memorized symbols and generalized symbols for the topic food. Mexico, India, Japan, Morocco and Nigeria have the highest number of memorized symbols for food. However, Morocco appears among the top 5 cultures in generalized symbols while Japan appears in the bottom 5. Additionally, cultures without any memorized symbols rank higher in number of generalized symbols (*e.g.* Yemenis for clothing and Tribagonian for food). Cultures such as these where the model wasn't able to memorize anything, prompts the need for generalizations in the next token sampling process.

For symbols that partially contain or are a combination of symbols from diffuse association, we find that they are generalizations which can be traced to high-frequency symbols resulting from diffuse association. These comprise of about 0.1% and 0.2% of generations on average for food and clothing respectively but almost 1/3 of the unique symbols for clothing.

## 4.6 THE DISTRIBUTION OF FOUR TYPES OF CULTURE-SYMBOL ASSOCIATIONS

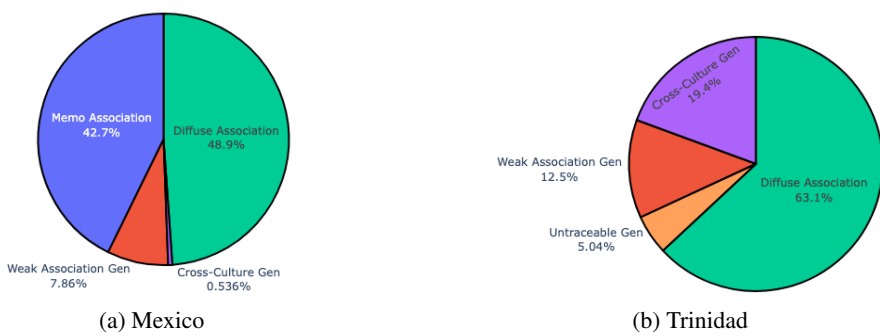

(a) Mexico                    (b) Trinidad

Figure 7: While some cultures contain no memorized association in their generations (Fig7b), cultures like Mexico's almost 1/2 generations comprise of memorized association (Fig 7a)

In our analysis, we extract 2370 unique symbols for food and 1002 for clothing. Of these, 4.1% (98 symbols) and 10.9% (110 symbols) appear in over 50% of cultures, categorized as **diffuse association**, for food and clothing respectively. For food, 46.12% (1098 symbols) are identified as memorized association, and 31.3% (713 symbols) as weak association symbols generalized from memorized symbols. In contrast, for clothing, 25.78% (258 symbols) are memorized association, and 31.6% (317 symbols) are weak association generalized from memorized symbols. Additionally, a smaller fraction of food symbols (7.6%, or 180 symbols) and a significant portion of clothing symbols (nearly one-third, or 332 symbols) are **weak association generalized from high-frequency symbols of diffuse association**. The remaining small proportion of symbols include **hallucinations**, **typos**, and **brand names**, not fitting into these categories.

While diffuse association only comprise of a small proportion of the total unique symbols extracted from responses, they comprise a **significant proportion** (91.12% for clothing and 79.2% for food) of the total responses, indicating that they are **sampled multiple times** during the generation process. Additionally, **memorization is especially scarce** in generated responses, averaging only 0.76% for clothing and 4.12% for food while traceable generalization averages to 3.1% and 4.9% for both topics respectively. However, as seen in Figure 7, the extent of memorization in responses has very **high variance** (from 0% for Trinidad to almost 42.2% for Mexico in food). Cross-culture generalization, while only averaging 4% and 11% respectively for clothing and food, exhibits high variance with cultures with a high number of memorized symbols having lesser cases of generating symbols memorized for other cultures. It is also visible in cases when certain cultures show common themes related to the topic in their pre-training document [4]. More analysis can be found in Appendix H.2

## 5 CONCLUSION

In conclusion, our work introduces MEMOED, a framework for attributing culture-conditioned generations of language models to memorization from pretraining data. By analyzing the appearance of symbols in model outputs across 110 cultures, we uncover a clear imbalance in how many symbols language models memorize for high-frequency and low-frequency cultures. In addition, models tend to prioritize generating high-frequency symbols that are not specific to any culture over memorized symbols, while also struggling to generalize from memorized cultural symbols with lower prevalence in the pretraining data. This highlights significant limitations in current pretraining processes, where models prefer frequently occurring, diffusely associated symbols at the expense of diverse, culture-specific knowledge. Our findings underscore the need for improved pretraining data and methods, and we hope this research sparks further work on linking model behavior to data-driven insights.

---

[4]As shown through keywords in Table 3

Acknowledgments

This research is supported in part by the Office of the Director of National Intelligence (ODNI), Intelligence Advanced Research Projects Activity (IARPA), via the HIATUS Program contract #2022-22072200006, the Defense Advanced Research Projects Agency with award HR00112220046, and NSF IIS 2048211. The views and conclusions contained herein are those of the authors and should not be interpreted as necessarily representing the official policies, either expressed or implied, of ODNI, IARPA, or the U.S. Government. We would like to thank all the collaborators in USC INK research lab for their constructive feedback on the work.

## LIMITATIONS

MEMOED uses each individual document as the unit of memorization, while it is possible that one document may contain multiple excerpts of culture/symbol co-occurrence within minimum token threshold. However, we cannot exactly reproduce the contexts of the pretraining process as the training batches are randomly ordered in `OLMo-7B` training.

Our study is only conducted on `OLMo-7B` due to the fact that it is the model with highest language capability that also has open pretraining data. How our conclusions may hold for non-OLMo family models is unknown; however, our methodology introduced in §3 is transferrable for analyzing any model, as long as their pretraining data is accessible.

## REPRODUCIBILITY STATEMENT

**Algorithm.** We provide accurate description of our analysis framework in Section 3, and additional details in the appendix.

**Prompt Engineering.** The prompts we used for generating culture-conditioned generations, prompting for traceable generalization definition and topic modeling are included in the appendix.

**Data and Source Code.** Data and source code will be released upon acceptance.

**Crowdsourcing.** Instructions for Prolific annotators are available in Appendix E.

## ETHICS STATEMENT

**Data.** All data we collected through LLMs in our work are released publicly for usage and have been duly scrutinized by the authors. Data for all human studies that we conduct are also publicly released with this work, with appropriate annotator anonymizations.

**Crowdsourcing.** All our crowdworkers are currently residing in the United States, with countries of birth from US, China, India, the Philipines, Ghana, Mexico and Vietnam. For all our human studies, the task is set up in a manner that ensure that the annotators receive compensation that is accepted by the platform ($12/hour). Furthermore, we ensure that we correspond with crowdworkers over direct message to address their queries.

**Potential Use.** Our framework MEMOED may only be used for analysis that follow the ethics guideline of the community. Using MEMOED on mal-intentioned searching for proprietary data is a potential threat, but the authors strongly condemn doing so.

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

## A  110 CULTURES

| Geographic Region | Countries and Regions |
|---|---|
| Eastern-European | Albania, Armenia, Belarus, Bosnia and Herzegovina, Bulgaria, Croatia, Czechia, Georgia, Greece, Hungary, Kosovo, Moldova, Montenegro, North Macedonia, Poland, Romania, Russia, Serbia, Slovakia, Slovenia, Turkey, Ukraine |
| African-Islamic | Algeria, Egypt, Ethiopia, Ghana, Kenya, Libya, Morocco, Nigeria, Rwanda, Tunisia, Zambia, Zimbabwe |
| Western-European | Andorra, Austria, Belgium, Finland, France, Germany, Ireland, Italy, Luxembourg, Netherlands, Portugal, Spain, Switzerland, United Kingdom |
| Latin-American | Argentina, Bolivia, Brazil, Chile, Colombia, Dominican Republic, Ecuador, El Salvador, Guatemala, Haiti, Mexico, Nicaragua, Peru, Puerto Rico, Uruguay, Venezuela |
| English Speaking | Australia, Canada, New Zealand, Trinidad and Tobago, United States, South Africa |
| Central-Asian | Azerbaijan, Kazakhstan, Kyrgyzstan, Mongolia, Tajikistan, Uzbekistan |
| South-Asian | Bangladesh, India, Maldives, Pakistan |
| Baltic | Estonia, Latvia, Lithuania |
| Nordic | Denmark, Finland, Iceland, Norway, Sweden |
| East-Asian | China, Hong Kong, Japan, Macau, South Korea, Taiwan |
| Southeast-Asian | Indonesia, Malaysia, Myanmar, Philippines, Singapore, Thailand, Vietnam |
| Middle-Eastern | Cyprus, Iran, Jordan, Lebanon, Palestine, Kuwait, Qatar, Saudi Arabia, Yemen |

Table 4: Countries and Regions for each geographic region, according to  (Haerpfer & Kizilova, 2012).

```
Instructions:
    ● Be helpful and answer questions concisely. If you don't know the
      answer, say 'None'
    ● Utilize only the tokens in the given sentence for generating
      phrases/words for the given query.
    ● Incorporate your preexisting knowledge to enhance the depth and
      relevance of your response.
    ● Cite your sources when providing factual information.

I had a document on which I ran Latent Dirichlet Allocation and I got
the following outputs:

<lda_output>

From this LDA output, list almost 5 possible topics which can be
modeled from the document.
A topic is a phrase or a word denoting a theme and you can infer it
from words or probabilities given above.
Do not provide any explanation or description for the topics you
generate. If you do not know more topics, stop and say DONE.
```

Figure 8: Prompt for `LLAMA-3.1-8B` in Topic Modeling Pipeline

## B  TOPIC MODELING

### B.1  METHODOLOGY

For any culture $C$ and its set of memorized symbols $m(C)$, we select a symbol $S \in m(C)$ and identify the set of cultures $C'_G$ which also generated $S$ but not through a memorization. For each culture $C' \in C'_G$ and for $C$, we retrieve pre-training documents where the two cultures co-occur, forming a set $D^{cc'}$. We apply the metrics defined in Section 3.3 to filter these documents, obtaining a subset $D^{cc'}_r \subseteq D^{cc'}$ that are relevant to the association of the two cultures. We further refine $D^{cc'}_r$ by removing documents that do not contain the symbol $S$, resulting in a final set $D^{cc's}_r$, which is relevant to the association between cultures $C$ and $C'$ and contains the memorized symbol $S$.

Subsequently, we use a sliding window of size $2048$ to create chunks from each document $d \in D^{cc's}_r$. We employ Latent Dirichlet Allocation (LDA) (Blei et al., 2003) to model five topics from each set of chunks corresponding to a document. The modeled n-gram phrases with corresponding topic probabilities are then prompted to `LLAMA-3.1-8B-Instruct` (Dubey et al., 2024) The LLM generates interpretable n-gram topic phrases, which are then filtered for repetitions using cosine similarity scores calculated with `XLM-RoBERTa-large` embeddings (Conneau, 2019). Finally, we extract the top five keywords from these topics using TF-IDF.

### B.2  PROMPT

In figure 8, we provide the prompt used for prompting `LLAMA-3.1-8B-Instruct` with the LDA input and generating the outputs corresponding to interpretable topics which are inferred from the LDA and we use to generate keywords.

In Table 5, we extend our study of pre-training documents (Table 3) pertaining to cross-cultural generalization from one culture to another for more cases of cultures which generate these memorized symbols with a lower count of relevant documents than the cultures discussed before. We notice suprisingly similiar themes in the pre-training documents such as the discussion around "religion" in documents where Hijab, Iran and any culture X co-occur. For Kimono and Japan, we notice a similar common theme surrounding "fashion". We hypothesize that such common themes also

| Symbol | Mem. Culture | Non-Mem. Culture | Topic Modelling Keywords |
|---|---|---|---|
| Hijab | Iran | Iraq | `[woman, government, islamic, war, politics, kurdish, people, conflict, protest, muslim]` |
| Hijab | Iran | Pakistan | `[woman, muslim, islamic, women, hijab, issues, government, rights, people, culture]` |
| Hijab | Iran | Indonesia | `[woman, islamic, muslim, hijab, fashion, law, women, islam, culture, government]` |
| Hijab | Iran | Egypt | `[woman, muslim, islamic, women, islam, arab, government, hijab, culture, politics]` |
| Kimono | Japan | Italy | `[fashion, art, tokyo, culture, design, food, hotel, experience, clothing, travel]` |
| Kimono | Japan | Kenya | `[travel, fashion, art, experience, culture, africa, african, design, food, names]` |
| Kimono | Japan | El Salvador | `[arts, martial, blue, color, dell, laptop, ryu, asian, friends, indigo]` |
| Biryani | India | Bangladesh | `[food, restaurant, dish, recipe, dishes, cuisine, cooking, bengali, chicken, recipes]` |
| Biryani | India | Singapore | `[food, restaurant, dish, cuisine, dishes, experience, biryani, chicken, cooking, options]` |
| Biryani | India | Iran | `[food, biryani, restaurant, cuisine, dish, recipe, saffron, hyderabad, spice, cooking]` |
| Churrasco | Brazil | Peru | `[food, restaurant, experience, cuisine, bar, dining, sushi, dish, london, city]` |
| Churrasco | Brazil | Colombia | `[food, restaurant, latin, bride, dating, beach, cheese, recipe, meat, travel]` |
| Churrasco | Brazil | Uruguay | `[food, meat, restaurant, cuisine, rio, american, bbq, experience, south, dish]` |

Table 5: Keywords modeled from pretraining documents in cases of cross-culture generalization (continued)

| Topic | Prompt Template |
|---|---|
| food | [instruction]. [symbol] is probably a [culture] food item which is a type of |
| clothing | [instruction]. [symbol] is probably a [culture] clothing item which looks like a type of |

Table 6: Prompts for generating definitions of memorized symbols to detect Weak-Association Generalization

cause models to overmemorize and generate memorizations from one culture into another and not necessarily when the model is devoid of memorizations in the next token space (which would be the case for cultures which have no memorizations).

## C  PROMPTS

### C.1  WEAK-ASSOCIATION GENERALIZATION PROMPT

[instruction] in Table 6 refers to the instruction (note that any typos in the prompt are intentional and included for the purpose of jailbreaking):

```
You are a helpful agent who is knewledgable about celtures from
all places.  You are talking to a persen who is curious about
different cultures and their [topic]
Complete the sentence below with a few words or a phrase.  If you
do not know the answer, print don't know.
```

## D    COMPUTING MINIMUM DISTANCE B/W TWO N-GRAMS

This section elaborates on the algorithm employed by us for computing minimum distance between two n-grams in a pretraining document and reporting the $d_{TOK}(C, S, D)$ metric. It calculates the context length difference between the n-grams $C$ and $S$, as observed by the LLM during pre-training. We hypothesize that for a pre-trained language model with a sequence length $L$, a smaller $d_{TOK}(C, S, D)$ indicates more frequent co-occurrence of the two n-grams across training batches. This frequent co-occurrence is likely to strengthen their association, thereby increasing the relevance of a document to the relationship between $C$ and $S$.

The algorithm described in Algorithm 1 computes the minimum token distance between two n-grams within a text, using a tokenizer to process the input and mark relevant tokens. Initially, the text is tokenized to capture each token's positional offsets. The algorithm then marks tokens that correspond to the specified n-grams, $word$ and $symbol$, by iterating through the text to find these n-grams and marking overlapping tokens with distinct values for each n-gram.

Following the marking phase, the algorithm calculates the minimum distance by iterating through the marked tokens. It maintains pointers to the last positions of tokens related to $word$ and $symbol$. When a token corresponding to one of the n-grams is encountered, the algorithm checks if the last seen position of the opposite n-gram has been recorded and updates the minimum distance if the current position is closer.

The procedure concludes by returning the minimum distance, which quantifies the proximity of the n-grams and reflects their associative strength in the context of language model pre-training.

## E    HUMAN ANNOTATION SETUP USING PROLIFIC

We designed a human annotation task using Google Forms, automatically populated via Google Apps Script with symbols related to food and clothing from eight different cultures. Figure 9 provides an overview of the form setup, while Figure 10 shows an example of a question where participants were asked to evaluate whether a specific food is a cultural food item of some culture. Annotators were required to select the most appropriate classification based on their knowledge of the culture in question. This process enabled us to collect reliable data regarding culturally emblematic food and clothing items.

## F    ABLATION STUDY

### F.1    ABLATION ON HYPERPARAMETERS

In the original design of our decoding process, multinomial sampling was employed with a set of specified hyperparameters: *temperature=1.0*, *top_p=0.95*, *top_k=50*, *max_tokens=30*, and *num_return_sequences=100*. The stopping criterion was established as the period ('.') character. To explore the impact of these parameters on the generation results, an ablation study was conducted where *top_k* values of 20 and 80, and *temperature* values of 0.75 and 1.25 were tested against the original settings. We observed an overlap coefficient of greater than 90% in all the four cases. This tells us that the sampling conditions did not cause or change our findings.

### F.2    ABLATION ON OLMo-7B VARIANTS

In order to verify that conclusions we find on OLMo-7B hold on other modalities, we reproduce some of the experiments on a newer variant of OLMo-7B, OLMo-7B-0424. We collect culture-conditioned generations for both food and clothing on OLMo-7B-0424, which is trained on Dolma

---

**Algorithm 1** Calculate minimum token distance between two n-grams

---

1: **procedure** MINTOKENDISTANCE($text$, $word$, $symbol$, $tokenizer$)
2:   $encoding \leftarrow tokenizer(text$, return_offsets_mapping=True$)$
3:   $tokens \leftarrow encoding.tokens()$
4:   $token\_offsets \leftarrow encoding['offset\_mapping']$
5:   $marks \leftarrow [0] * len(tokens)$
                                   ▷ Mark tokens corresponding to symbol and word
6:   **for** $phrase \in \{symbol, word\}$ **do**
7:     **for** $start$ in $text$ **do**
8:       **if** $text.find(phrase, start) \neq -1$ **then**
9:         $end \leftarrow start + len(phrase)$
10:         **for** $i, (s, e)$ in enumerate $token\_offsets$ **do**
11:           **if** $s \neq None \wedge e \neq None \wedge s < end \wedge e > start$ **then**
12:             $marks[i] \leftarrow \max(marks[i]$, if $phrase = symbol$ then 2 else 1$)$
13:         $start \leftarrow end$
14:   $min\_distance \leftarrow \infty$
15:   $last\_symbol \leftarrow -1$
16:   $last\_word \leftarrow -1$
                                   ▷ Compute minimum distance between marked tokens
17:   **for** $i$ from 0 to $len(marks)$ **do**
18:     **if** $marks[i] = 2$ **then**
19:       $last\_symbol \leftarrow i$
20:       **if** $last\_word \neq -1$ **then**
21:         $min\_distance \leftarrow \min(min\_distance, i - last\_word)$
22:     **else if** $marks[i] = 1$ **then**
23:       $last\_word \leftarrow i$
24:       **if** $last\_symbol \neq -1$ **then**
25:         $min\_distance \leftarrow \min(min\_distance, i - last\_symbol)$
26:   **return** $min\_distance$

---

| Ordering | Food | w/ Clothing |
|---|---|---|
| From Top 10 (↑) | Morocco - 107
Bangladesh - 99
Iceland - 99
Sweden - 96
Ethiopia - 90 | Azerbaijan - 97
Bolivia - 96
Chile - 91
India - 76
Kenya - 74 |
| From Bottom 10 (↓) | France - 42
Singapore - 42
Britain - 38
Indonesia - 36
Australia - 35 | Germany - 30
United States - 28
China - 26
Portugal - 24
France - 21 |

Table 7: Cultures chosen for ablating on `OLMo-7B-0424` and their corresponding number of unique symbols

**Instructions**

In this task, we ask you to classify 11 food items as whether it is a "cultural food item" of the American culture. A "cultural food item" is commonly recognized as either originating from the American culture or emblematic to some religion/ethnic group/community within the American culture.

We ask that you classify each item into one of the five options below, based on your personal experience and knowledge about the American culture:

1. I know this food item and it is a cultural food item of American culture
2. I know this food item but it is not a cultural food item of American culture
3. This food item is a typo but it is a cultural food item of American culture
4. This food item is a typo and it is not a cultural food item of American culture
5. I don't know this food item

**Additional Instructions**

If you see "null" as a symbol in the question, please select option 5 "I don't know this food item".
\*\*IMPORTANT: Some questions in the form are included as attention checks. If you fail any of these, you'll need to return the questionnaire, and you won't be eligible for a reward.\*\*

**Sample Annotations**

\*\*Please refer to sample annotations below to understand the criteria for each option.\*\*
(Assume you are from Malaysia)

Q1: Is "Ayam Goreng" a cultural food item of the Malaysian culture? (Note: Ayam Goreng is a type of fried chicken commonly eaten in Indonesian and Malaysian cultures)

A1: (Select Option 1) I know this food item and it is a cultural food item of the Malaysian culture. Explanation: Ayam Goreng is emblematic only to Malaysian and Indonesian cultures, but not other cultures.

Q2: Is "fried chicken" a cultural food item of the Malaysian culture?

A2: (Select Option 2) I know this food item and it is not a cultural food item of the Malaysian culture. Explanation: Although there is a type of fried chicken emblematic to Malaysian culture, fried chicken is eaten in many other cultures, so "fried chicken" does not qualify as a cultural food item.

Q3: Is "Sushi" a cultural food item of the Malaysian culture? (Note: Sushi is a type of Japanese dish with vegetables, meat/seafood and seaweed wrapped around rice)

A3: (Both Option 2 and 5 are correct) I know this food item and it is not a cultural food item of the Malaysian culture / I don't know this food item. Explanation: depending on your knowledge about specific food items, you can select either options.

Q4: Is "rice" a cultural food item of the Malaysian culture?

A4: (Select Option 2) I know this food item and it is not a cultural food item of the Malaysian culture. Explanation: Rice is prevalent globally eaten by almost all cultures all around the world, and therefore it does not qualify as a cultural food item.

Figure 9: Example of Google Form Used for Cultural Food Annotation

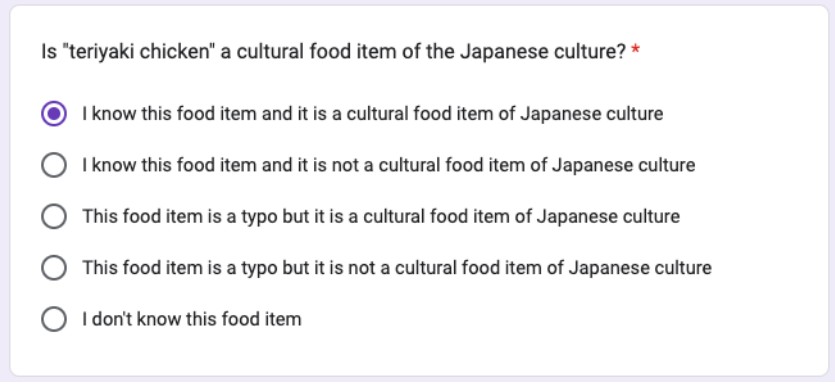

Figure 10: Sample Question from Google Form on Cultural Food Classification

1.7. Although `OLMo-7B-0424`is the same model family as `OLMo-7B`, `Dolma` 1.7 contains pre-training documents that are not in `Dolma` 1.5, and `OLMo-7B-0424` is trained with an updated algorithm from `OLMo-7B`. Other models supported by the WIMBD API, such as `Pythia` (Biderman et al., 2023), are not particularly capable of instruction following culture-conditioned generations, and therefore, analyzing their generations is less informative.

We only reproduce two main correlations in the main paper:

**The number of cultures a symbol with diffuse association is generated for and the number of pretraining documents it appears in (Section 4.3)** For `OLMo-7B-0424`, we obtain a moderate-to-strong correlation for both clothing (spearman $\rho = 0.507$, Kendall $\tau = 0.362$) and food (spearman $\rho = 0.416$, Kendall $\tau = 0.313$). Compared to `OLMo-7B` with clothing (spearman $\rho = 0.521$, Kendall $\tau = 0.367$) and food (spearman $\rho = 0.358$, Kendall $\tau = 0.260$), we see that even though the models and training data are different, the Spearman and Kendall correlations for food and clothing remain the same (both moderate-to-strong correlations). This means that the number of cultures a symbol with diffuse association was generated for and the number of pretraining documents it appears in is positively correlated, regardless of the model.

**The number of memorized symbols for a culture and the number of pretraining documents it appears in (Section 4.1)** For `OLMo-7B-0424`, we select 10 cultures out of 110, 5 from the 10 cultures with the highest number of unique symbols generated by `OLMo-7B-0424` and 5 from the 10 cultures with the lowest number of unique symbols generated by `OLMo-7B-0424`.

We obtain a moderate-to-strong correlation for both clothing (spearman $\rho = 0.591$, Kendall $\tau = 0.507$) and food (spearman $\rho = 0.829$, Kendall $\tau = 0.659$). Compared to `OLMo-7B` (on 110 cultures) with clothing (spearman $\rho = 0.540$, Kendall $\tau = 0.421$) and food (spearman $\rho = 0.670$, Kendall $\tau = 0.507$), we see that even though `OLMo-7B-0424` is tested on smaller number of cultures, for both clothing and food, the correlation of `OLMo-7B-0424`is more strongly positive. Therefore, the conclusion still holds that higher pretraining document counts of cultures increase the number of memorized symbols in culture-conditioned generations.

### F.3 ABLATION ON Z-SCORE FOR MEMOED

We study whether selecting a different z-score threshold would change the conclusions of MEM-OED on memorized symbols for all cultures. We perform an ablation study on setting the z-score to 2, which statistically means that the value is about 97.7 percentile. Empirically, a z-score below 2 does not indicate outliers, so we focus our ablation analysis only on cases where the z-score is 2.

When $z = 2$, we still get a moderate-to-strong correlation between 1) the number of memorized symbols for a culture and 2) the count of documents in which the culture appears in the pretraining corpora: for clothing, we obtain a spearman correlation of 0.569 and a Kendall correlation of 0.445; for food, we obtain a spearman correlation of 0.688 and a Kendall correlation of 0.519. This correlation is lower but similar to the original correlations found for z=2.6 (food: Spearman=0.670 and

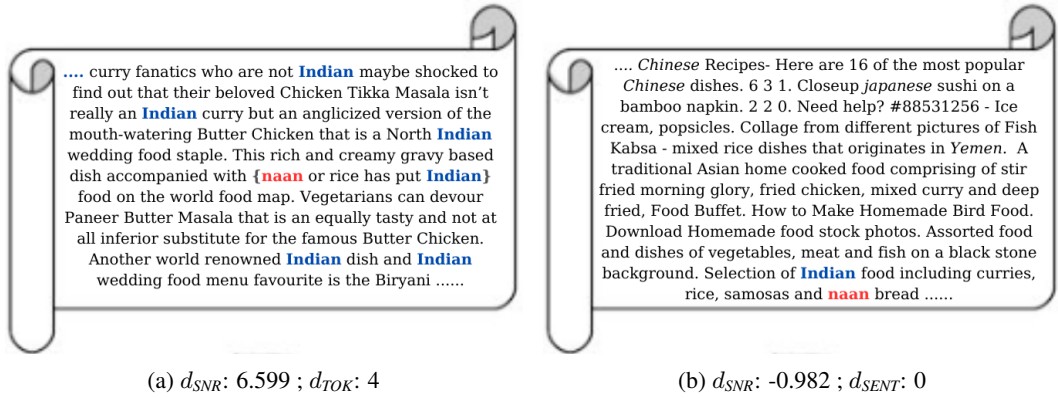

(a) $d_{SNR}$: 6.599 ; $d_{TOK}$: 4       (b) $d_{SNR}$: -0.982 ; $d_{SENT}$: 0

Figure 11: Examples of excerpts from relevant pretraining docs for Culture: "Indian" and Symbol: "Naan":

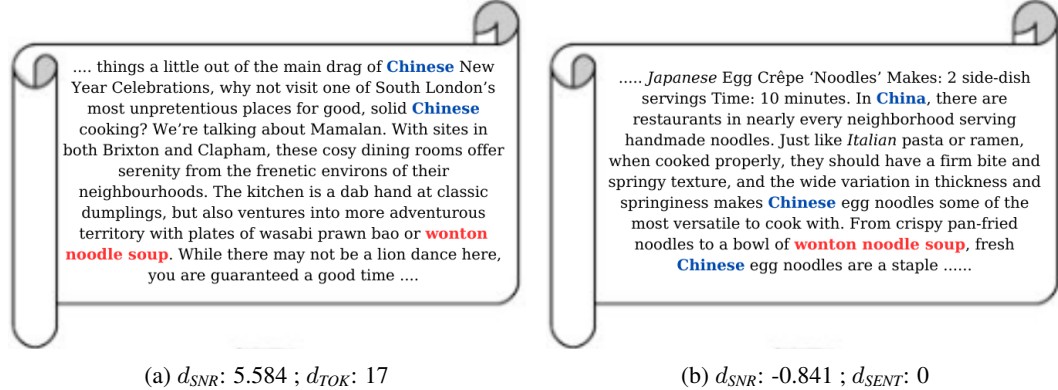

(a) $d_{SNR}$: 5.584 ; $d_{TOK}$: 17       (b) $d_{SNR}$: -0.841 ; $d_{SENT}$: 0

Figure 12: Examples of excerpts from relevant pretraining docs for Culture: "Chinese" and Symbol: "Wonton Noodle Soup":

Kendall=0.507; clothing: Spearman=0.540 and Kendall=0.421), showing that our conclusion on the relationship between a culture's memorized symbols and the culture's frequency in pretraining data is robust to a different z-score threshold.

In addition, we examine how lowering the z-score from 2.6 to 2 changes memorized symbols discovered for each culture. We compare each metric's agreement with human evaluation on clothing: when z=2.6, the weighted F1 score is 0.845, and when z=2, the weighted F1 score is 0.840. We can see that z = 2 has a slightly lower agreement with human categorization, suggesting that additional symbols that are marked as memorized symbols when z=2 are non-emblematic symbols according to human culture experts.

## G  TRAINING DOCUMENT EXCERPTS

In this section, we present excerpts from the pre-training documents classified as contributory to a culture-symbol association using MEMOED's $d_{SNR}$, $d_{TOK}$ and $d_{SENT}$ metrics.

In Figure 11, we present excerpts from two pretraining documents classified as contributory to the association between the culture: *Indian* and the symbol: *Naan*. We also report the relevant metric scores used to determine this. For Figure 11a, since the $d_{SNR}$ is greater than zero, the $d_{TOK}$ metric is used to ascertain the classification of this document. As visible in the excerpt, the culture "Indian" appears numerous times and in close proximity to the symbol "naan". Additionally, upon seeing the remaining part of the excerpt, we see that it is talking about Indian food items which indicates the relevancy of this document towards the association. On the other hand, for Figure 11b, since

| Culture | Memorized Symbols From | Pre-Training Count Rank (/110) |
|---|---|---|
| Trinbagonian | **American** (0.4%) | **101** |
| Macanese | **American** (0.5%) | **100** |
| Salvadoran | **American** (1%) | **99** |
| Zambian | **American** (0.6%) | **94** |
| Nicaraguan | **American** (0.4%) | 85 |
| Puertorriqueña | **American** (0.6%) | 70 |
| Egyptian | *Iranian* (2.9%) | 27 |
| Saudi | *Iranian* (6.2%) | 45 |
| Andorran | French (0.3%) | **110** |
| Hong Konger | French (0.6%) | 38 |

Table 8: Cultures Identified from Leave-One-Out-Correlation

the $d_{SNR}$ is between 0 and -1, we use the $d_{SENT}$ metric as explained in Section 3.3. We can observe similarly that although the ratio is less than zero, the document is not noisy and the local context is about Indian food item.

Similarly, in Figure 12, we present excerpts from two pretraining documents classified as contributory to the association between the culture: *Chinese* and the symbol: *Wonton Noodle Soup*. We can observe that the training document with a positive $d_{SNR}$ is not really talking about Chinese food items but rather talks about a prominent Chinese festival *i.e.* Chinese New Year and mentions the food delicacies being prepared then. Thus, through this it contributes to the association between the culture and symbol. On the other hand, for the document with negative $d_{SNR}$, we observe a relatively high concentration of cultural mentions in this excerpt and on a global level, the topic being discussed is restaurants in China when the food cultural symbol is mentioned. Hence we see how this document potentially contributes to the culture-symbol association.

# H ADDITIONAL RESULTS

## H.1 CROSS-CULTURE GENERALIZATION

| Topic | Keywords |
|---|---|
| food | food, foods, cuisine, cuisines, dish, dishes, meal, meals, recipe, recipes, menu, menus, breakfast, lunch, dinner, snack, snacks |
| clothing | clothing, clothes, apparel, garment, garments, outfit, outfits, attire, attires, dress, dresses, suit, suits, uniform, uniforms |

Table 9: Keyword list that we use to filter for topic-related pretraining documents.

To further evaluate cross-culture generalization across all 110 cultures, we obtain: (1) the percentage of a culture's responses that contain another culture's memorized symbols; (2) how often is a culture's memorized symbol generated for some other culture. Additionally, we calculate the correlation between each culture's metrics (1) and (2) with the frequency of topic-relevant occurrences of that culture in the pretraining corpora.

For (1), we observe a moderate negative correlation for food (Spearman $\rho = -0.521$, Kendall $\tau = -0.364$) indicating that cultures containing more memorized symbols from other cultures tend to occur less-frequently in food-related pretraining documents. We have shown this correlation using a scatter plot in Figure 13. However, for clothing, we observe a weak negative correlation (spearman $\rho = -0.099$, Kendall $\tau = -0.061$). To investigate this, we conduct a *leave-one-culture-out* experiment. In this analysis, we recalculated the correlations while systematically excluding one culture at a time. We then identify and list the top ten cultures causing the highest variation. Notably, these cultures are either those that predominantly contain generalization from regional cultures, such as *Egyptian* or *Saudi*, or those that are less frequently mentioned in the pretraining data, such as *Trinbagonian*. We have listed these ten cultures with the highest number of cross-

culture generalization in their responses, along with the culture whose memorized symbols appear the most in the former cultures, as well as their pretraining occurrence ranked out of all 110 cultures in Table 8. We observe that a majority of cultures have the highest generalization from *America* while Egypt and Saudi have a significant percentage of their generations memorized from one culture *i.e.* Iran.

For (2), our observations indicate that 34 cultures related to clothing and 86 related to food have their memorized symbols being generated at least once in other cultures' generations. Upon calculating correlations with these cultures, we observed moderate-to-high correlations for both clothing (Spearman $\rho = 0.763$, Kendall $\tau = 0.574$) and food (Spearman $\rho = 0.716$, Kendall $\tau = 0.531$). These results suggest that cultures whose symbols are frequently mentioned in other cultures' generations are also those more commonly appearing in topic-related pretraining documents. We show this correlation through scatter plots for both clothing and food in Figure 14.

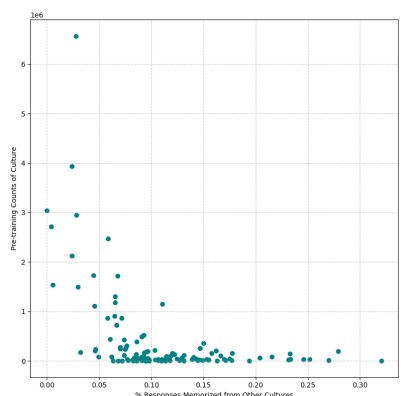

Figure 13: Correlation b/w number of memorized symbols from other cultures and pre-training counts for a culture

## H.2 RESULTS OVERVIEW

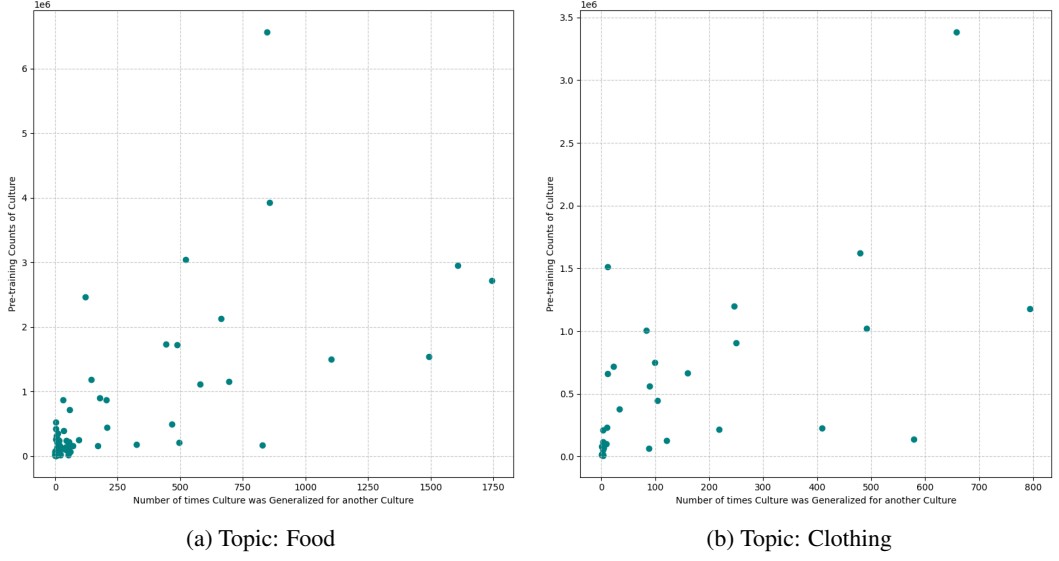

(a) Topic: Food

(b) Topic: Clothing

Figure 14: Cross-Culture Generalization

Continuing from Section 4.6, in this section we expand upon our findings and present some more results across the 110 cultures.

In Tables 10 and 11, we present the memorization and generalization statistics for food and clothing, respectively. Specifically, we provide the names of the top 5 and bottom 5 cultures, ranked by the percentage of their responses classified as either memorization or weak association generalization. Cultures with the highest percentage of memorized responses tend to correspond to those that appear more frequently in the pretraining dataset. However, notable exceptions exist, such as the culture *United States*, which, despite occurring frequently in the pretraining data and having a large number of memorized symbols, exhibits only 3.01% of its total responses as memorized, as shown in Figure 17a.

| Ordering | w/ Memorization | w/ Weak Association Gen. |
| --- | --- | --- |
| Top 5 (↑) | Mexico
India
Japanese
Morocco
Nigeria | Trinidad
Venezuela
South Korea
Morocco
Georgia |
| Bottom 5 (↓) | Qatar
South Africa
Tajikistan
Trinidad
Yemen | Germany
Japan
United States
Italy
Denmark |

Table 10: Memorization and Generalization Stats for Food

| Ordering | w/ Memorization | w/ Weak Association Gen. |
| --- | --- | --- |
| Top 5 (↑) | India
Saudi Arabia
Japan
Pakistan
Canada | Uruguay
Venezuela
Vietnam
Yemen
Zambia |
| Bottom 5 (↓) | Uruguay
Venezuela
Vietnam
Yemen
Zambia | Colombia
Peru
Nicargua
Venezuela
United States |

Table 11: Memorization and Generalization Stats for Clothing

We also observe that a culture with a high percentage of memorized responses does not necessarily have a large number of unique memorized symbols. For instance, Pakistan ranks 4th in memorization count for the topic of clothing but has relatively few unique memorized symbols. This indicates that for some cultures, `OLMo-7B` tends to repeatedly generate the same memorized symbols when sampled multiple times. Additionally, Table 11 shows that the bottom 5 cultures, which have the lowest percentage of their responses classified as memorized, exhibit the highest percentage of weak association generalization in their responses.

We further provide the distribution of additional cultures, similar to the analysis presented for Mexico and Trinidad in Section 4.6. Figure 15 illustrates the distribution of Chinese, Japanese, and Indian cultures for the topic of food. Notably, despite these three cultures being relatively high-frequency in the pretraining data, all exhibit very high percentage of symbols of diffuse association, exceeding 60% in each case. Interestingly, we also observe considerable variance in the overall presence of memorization, ranging from almost 30% for India to only 11.5% for China. Additionally, all three cultures exhibit a relatively low percentage of cross-culture generalization. This is likely due to their high frequency in the pretraining data, which results in their symbols being generalized to other less frequently occurring cultures.

In Figure 16, we compare the distributions of two less-frequently occurring cultures, *i.e.*, Myanmar and Yemen, for the topic of clothing. We observe that, apart from exhibiting very high rates of diffuse association symbols (greater than 70% in most cases), these cultures have no memorized symbols according to the classification provided by MEMOED. Yemen, in particular, demonstrates a notably high percentage of cross-culture generalization, approximately 21.1%.

Finally, in Figure 17, we present the distributions for the USA and Saudi Arabia within the topic of clothing. The results for the USA are particularly striking, as it is one of the most frequently occurring cultures in the pretraining dataset, yet nearly 96% of its responses consist solely of diffuse association symbols. Despite containing a substantial number of unique memorized symbols, only

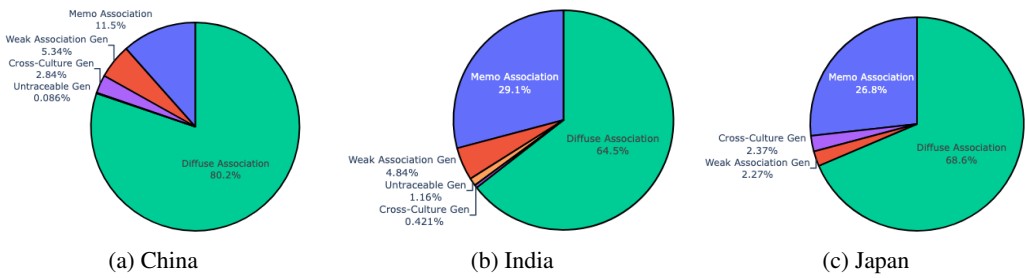

Figure 15: Distributions of China, India and Japan responses for Food

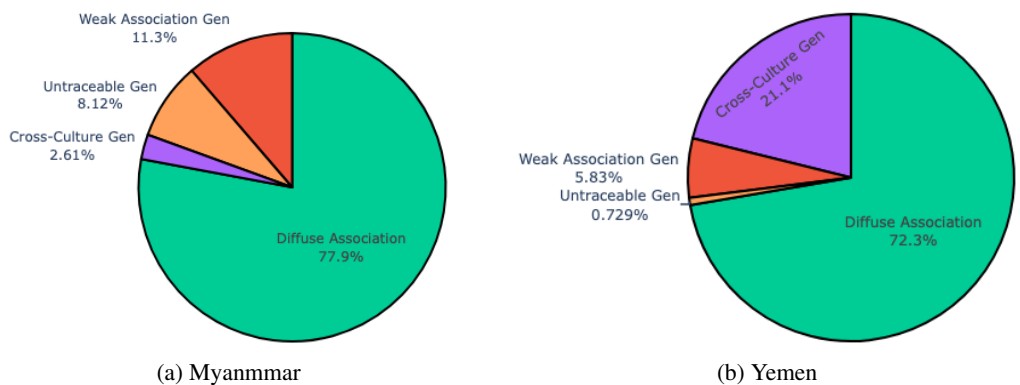

Figure 16: Clothing Stats - Mynammar and Yemen

3% of its responses qualify as memorization. In contrast, Saudi Arabia exhibits greater diversity, with significant percentages of both memorization and cross-culture generalization in its generated outputs.

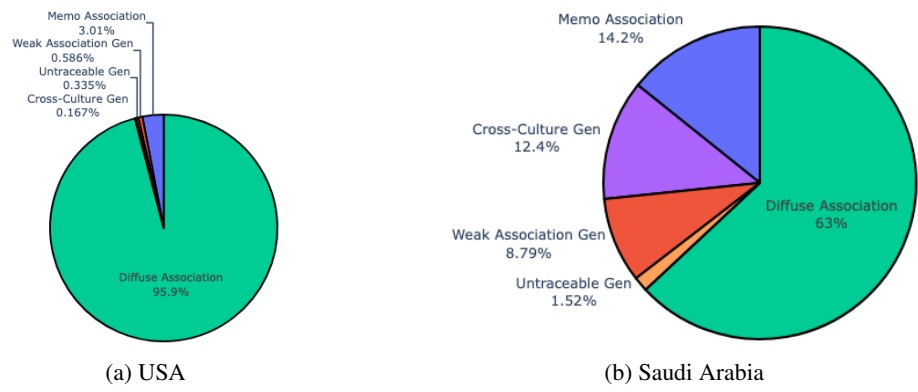

Figure 17: Clothing Stats - USA and Saudi Arabia

