# OpenReview forum: "Attributing Culture-Conditioned Generations to Pretraining Corpora"
_ICLR.cc/2025/Conference — ICLR 2025 Poster_

### Official Review · Reviewer_D6x9 · 2024-10-28

**Soundness:** 3
**Presentation:** 3
**Contribution:** 3
**Rating:** 8
**Confidence:** 2

**Summary:**

This paper introduces a novel framework called MEMOED, designed to analyze how pretraining data contributes to cultural biases in large language models (LLMs). The framework distinguishes between knowledge generated through memorization and generalization. By focusing on cultural topics such as food and clothing across 110 different cultures, the authors demonstrate that models tend to overmemorize symbols from highly represented cultures, while underperforming in generating culture-specific symbols for less represented ones. Through a detailed analysis of the OLMo-7B model, the paper offers a systematic method to trace how pretraining data influences model outputs, highlighting the limitations of current LLMs in producing diverse, culturally accurate generations and stressing the need for improved pretraining procedures to address these biases.

(Note: My review has been revised by an LLM for improved grammar.)

**Strengths:**

- The paper makes a significant contribution by addressing cultural biases in LLMs, and the MEMOED framework provides a valuable tool for tracking these biases.
- The study covers a broad scope, examining 110 cultures, which enhances the depth of the analysis.
- The concept of overmemorization introduced by the authors is intriguing and may have broader implications beyond cultural biases, potentially applying to other LLM phenomena.
- The paper opens up the possibility of examining cultural biases in multilingual LLMs across different languages, which could be an interesting direction for future research.

**Weaknesses:**

While the methodology appears sound, my only concern is the limited scope of the study, which focuses solely on the OLMo-7B model. As a result, the findings might be seen as a case study specific to this model. It would strengthen the paper if the authors included analyses for at least one additional LLM to broaden the generalizability of their findings.

**Questions:**

- Do the authors have any plans to propose methods for mitigating the cultural biases identified in this work?
- Comment: I recommend adjusting the notation for subscripts, such as $d_{TOK}$. It currently appears a bit unnatural, and applying italics to the subscript, such as d_{\textit{TOK}}, would enhance clarity.

---

> ### Author Response · Authors · 2024-11-22
> **Response to Review (1)**
>
> Thank you for your insightful review and very helpful comments! Below are our responses and we have updated the draft accordingly.
>
> >W1: While the methodology appears sound, my only concern is the limited scope of the study, which focuses solely on the OLMo-7B model. As a result, the findings might be seen as a case study specific to this model. It would strengthen the paper if the authors included analyses for at least one additional LLM to broaden the generalizability of their findings.
>
> We quickly collected culture-conditioned generations for both food and clothing on `olmo-7b-0424` (OLMo 1.7) which is trained on Dolma 1.7. Although this is the same model family as `OLMo-7B`, Dolma 1.7 contains pretraining documents not in Dolma 1.5, and OLMo 1.7 is trained with an updated algorithm from OLMo 7B. Other models supported by the WIMBD API we used, such as `Pythia`, are not particularly capable of instruction following culture-conditioned generations, and therefore, analyzing their generations is less informative.
>
> Due to the time constraints of the rebuttal, we are unable to completely reproduce all analysis done in the paper. We analyzed 2 correlations (added to Appendix E.2):
>
> 1) Between the number of cultures independent symbol was generated for and its pre-training count:
> | Model       | Category  | Spearman Correlation | Kendall Correlation |
> |-------------|-----------|----------------------|---------------------|
> | OLMo 1.7    | Food      | 0.416                | 0.313               |
> | OLMo 1.7    | Clothing  | 0.507                | 0.362               |
> | OLMo 7B     | Food      | 0.358                | 0.260               |
> | OLMo 7B     | Clothing  | 0.521                | 0.367               |
>
> We see that even though models and training data are different, the Spearman and Kendall correlations for food and clothing remain the same (both moderate-to-strong correlations). This means that the number of cultures an independent symbol was generated for and its pretraining count is positively correlated, regardless of the model.
>
> 2) Between number of memorized symbols and pre-training counts (OLMo 1.7 is sampled on 10 cultures due to time constraint, detail see Appendix E.2):
>
> | Model       | Category  | Spearman Correlation | Kendall Correlation |
> |-------------|-----------|----------------------|---------------------|
> | OLMo 1.7    | Food      | 0.829                | 0.659               |
> | OLMo 1.7    | Clothing  | 0.591                | 0.507               |
> | OLMo 7B     | Food      | 0.670                | 0.507               |
> | OLMo 7B     | Clothing  | 0.540                | 0.421               |
>
> For this experiment, exhaustively searching for all memorized symbols of all 110 cultures requires running MEMOed on all symbol-culture pairs, which is not feasible due to the rebuttal time constraint. Nonetheless, we see that for clothing and food, the correlation of Olmo 1.7 increases despite the small number of cultures we evaluated. Therefore, the conclusion still holds that higher pretraining counts of cultures increase the number of memorized symbols in culture-conditioned generations.

---

> > ### Author Response · Authors · 2024-11-22
> > **Response to Review (2)**
> >
> > > Q1: Do the authors have any plans to propose methods for mitigating the cultural biases identified in this work?
> >
> > We are going to add a discussion paragraph on how our findings can assist in existing unlearning or bias mitigation approaches for solving cultural biases in culture-conditioned generations, and we write our thoughts below:
> >
> > Many works have been done on unlearning knowledge, mitigating biases, or combatting spurious patterns through influential data [1][2][3][4][5]. These works follow the pipeline of influential pattern attribution, mostly through influence function [6] or feature attribution [7], and then demote the patterns by either learning confound-invariant representations, e.g., adversarial training [8] or data-based approaches to balance the training data, e.g., counterfactual data augmentation [9]. Our discovery of symbol overmemorization can be an alternative to the first step, as it helps identify high-frequency symbols that are biasing model generations, which are the targets of demotions in the second step. Our discovery of cultures with inadequate memorized symbols can also facilitate strategies for data augmentation: from pretraining documents containing these cultures, we find symbols that are related to the culture but not generated by the models, and these will be the symbols that need augmentation.
> >
> > [1] Mitigating Social Biases in Language Models through Unlearning
> >
> > [2] Influence Tuning: Demoting Spurious Correlations via Instance Attribution and Instance-Driven Updates
> >
> > [3] Adversarial Unlearning: Reducing Confidence Along Adversarial Directions
> >
> > [4] Debiasing Machine Unlearning with Counterfactual Examples
> >
> > [5] Unlearn What You Want to Forget: Efficient Unlearning for LLMs
> >
> > [6] Understanding black-box predictions via influence functions
> >
> > [7] “Why should I trust you?”: Explaining the predictions of any classifier
> >
> > [8] Adversarial removal of demographic attributes from text data
> >
> > [9] Counterfactual data augmentation for mitigating gender stereotypes in languages with rich morphology

---

> > > ### Comment · Reviewer_D6x9 · 2024-11-23
> > >
> > > Thank you for submitting the response, I am satisfied with the response and revised my rating accordingly.

---

> > > > ### Author Response · Authors · 2024-11-25
> > > > **Thank you for your feedback!**
> > > >
> > > > Dear Reviewer D6x9,
> > > >
> > > > Thank you for your positive feedback! We are happy to hear that our rebuttal addressed your concerns.
> > > >
> > > > Sincerely,
> > > >
> > > > Paper13342 Authors

---

### Official Review · Reviewer_UeFD · 2024-11-03

**Soundness:** 3
**Presentation:** 3
**Contribution:** 3
**Rating:** 8
**Confidence:** 4

**Summary:**

This paper introduces a novel symbol attribution framework to determine whether symbols in LLM generations, conditioned on a culture, result from memorization of pretraining data. The authors' thorough analysis shows that high-frequency symbols are easily memorized but independent of any culture regardless of their correctness. Additionally, by showing the imbalance between the memorization of high-frequency and low-frequency cultural symbols, this paper underscores the need for improved pretraining data and methods to mitigate cultural biases.

**Strengths:**

**S1**. This paper introduces a novel symbol attribution framework to determine whether the symbols in LLM generations, conditioned on a culture, result from memorization of pretraining data.

**S2**. The authors provide a nuanced categorization of symbols based on their memorization/generalization levels and a thorough analysis of their relationship with the pretraining data.

**S3**. Their findings demonstrate how LLMs fail to represent cultures that are low-frequency in the pretraining data, calling for improved pretraining data and methods.

**Weaknesses:**

**W1**. The study is limited to only one pretraining corpus and one LLM, which is understandable given the scarcity of open resources.

**W2**. This study relies on searching for symbols in culture-conditioned generations within the pretraining data and provides a relational analysis. However, it does not guarantee that the selected training documents are causally decisive for the symbols in question. Incorporating influence functions [1] could provide insights into causal relationships. While the computational cost might be an issue, they could be applied to a subset of the dataset or specific experiments.

**W3**. Lines 427-430 require further explanation. What do these correlations imply?

**W4**.  While this study highlights the existing problems with underrepresented cultures in pretraining corpora from a new perspective, it fails to address or propose potential directions for solving these issues. Without this, the paper remains another verification of known problems, which is still valuable but not particularly groundbreaking. The authors should discuss how their findings could inform improved pretraining data/methods or mitigation strategies that do not require changes in pretraining.

**References**
1. Studying Large Language Model Generalization with Influence Functions, Grosse et al., 2023

**Questions:**

**Q1**. In Figure 1, the top-down order does not match the numerical order. Why are memorized symbols shown at the bottom?

**Q2**. In Figure 3, what does "overgeneralization" refer to? It is not mentioned in the text. Do you mean "overmemorization" instead? The same applies to the caption of Table 3.

**Q3**. How are culture-referring n-grams defined for the Document-Signal to Noise Ratio?

**Q4**. Why do the memorization classification criteria differ for cases where n(C_G) > 5 and n(C_G) < 5?

**Q5**. What do the bold texts represent in the "topic modeling keywords" column of Table 3?


**Typos**:

- Line 345: "none-memorized" should be "non-memorized."
- Lines 106-107: "for less prevalent symbols" -> "for less prevalent cultures."
- There is inconsistent use of "memorisation" and "memorization" throughout the text. It would be better to use one consistently.

---

> ### Author Response · Authors · 2024-11-22
> **Response to Review (1)**
>
> Thank you for your insightful review and very helpful comments! Below are our responses and we have updated the draft accordingly.
>
> > W1. The study is limited to only one pretraining corpus and one LLM, which is understandable given the scarcity of open resources.
>
> We quickly collected culture-conditioned generations for both food and clothing on `olmo-7b-0424` (OLMo 1.7) which is trained on Dolma 1.7. Although this is the same model family as `OLMo-7B`, Dolma 1.7 contains pretraining documents not in Dolma 1.5, and OLMo 1.7 is trained with an updated algorithm from OLMo 7B. Other models supported by the WIMBD API we used, such as `Pythia`, are not particularly capable of instruction following culture-conditioned generations, and therefore, analyzing their generations is less informative.
>
> Due to the time constraints of the rebuttal, we are unable to completely reproduce all analysis done in the paper. We analyzed 2 correlations (added to Appendix E.2):
>
> 1) Between the number of cultures independent symbol was generated for and its pre-training count:
> | Model       | Category  | Spearman Correlation | Kendall Correlation |
> |-------------|-----------|----------------------|---------------------|
> | OLMo 1.7    | Food      | 0.416                | 0.313               |
> | OLMo 1.7    | Clothing  | 0.507                | 0.362               |
> | OLMo 7B     | Food      | 0.358                | 0.260               |
> | OLMo 7B     | Clothing  | 0.521                | 0.367               |
>
> We see that even though models and training data are different, the Spearman and Kendall correlations for food and clothing remain the same (both moderate-to-strong correlations). This means that the number of cultures an independent symbol was generated for and its pretraining count is positively correlated, regardless of the model.
>
> 2) Between number of memorized symbols and pre-training counts (OLMo 1.7 is sampled on 10 cultures due to time constraint, detail see Appendix E.2):
>
> | Model       | Category  | Spearman Correlation | Kendall Correlation |
> |-------------|-----------|----------------------|---------------------|
> | OLMo 1.7    | Food      | 0.829                | 0.659               |
> | OLMo 1.7    | Clothing  | 0.591                | 0.507               |
> | OLMo 7B     | Food      | 0.670                | 0.507               |
> | OLMo 7B     | Clothing  | 0.540                | 0.421               |
>
> For this experiment, exhaustively searching for all memorized symbols of all 110 cultures requires running MEMOed on all symbol-culture pairs, which is not feasible due to the rebuttal time constraint. Nonetheless, we see that for clothing and food, the correlation of Olmo 1.7 increases despite the small number of cultures we evaluated. Therefore, the conclusion still holds that higher pretraining counts of cultures increase the number of memorized symbols in culture-conditioned generations.
>
> > W2. This study relies on searching for symbols in culture-conditioned generations within the pretraining data and provides a relational analysis. However, it does not guarantee that the selected training documents are causally decisive for the symbols in question. Incorporating influence functions could provide insights into causal relationships. While the computational cost might be an issue, they could be applied to a subset of the dataset or specific experiments.
>
> Thank you for your suggestion. Currently we are running experiments on influence functions on pretraining documents and will update the results in a few days.
>
> > W3. Lines 427-430 require further explanation. What do these correlations imply?
>
> Thank you for pointing this out. We found an error in the calculation, and here are the updated correlations:
>
> - Clothing: Spearman Correlation: 0.551; Kendall Correlation: 0.385
> - Food: Spearman Correlation: 0.519; Kendall Correlation: 0.380
>
> We see a moderate to strong positive correlation between 1) the average over ratios of the document count of an independent symbol to the document count of every memorized symbol and 2) the number of cultures that the independent symbol is generated for. This suggests that the higher frequency the independent symbol appears in pretraining data, the more cultures the symbol is generated for. This result shows the reason why an independent symbol, which is not associated with any culture in pretraining data, is generated for that culture is due to the symbol itself having a very high frequency in pretraining data.

---

> ### Author Response · Authors · 2024-11-22
> **Response to Review (2)**
>
> > W4. While this study highlights the existing problems with underrepresented cultures in pretraining corpora from a new perspective, it fails to address or propose potential directions for solving these issues. Without this, the paper remains another verification of known problems, which is still valuable but not particularly groundbreaking. The authors should discuss how their findings could inform improved pretraining data/methods or mitigation strategies that do not require changes in pretraining.
>
> We politely disagree that our paper is “another verification of known problems.” While our paper did highlight the problems with underrepresented cultures in pretraining corpora, we also **introduced a framework for identifying cultures with insufficient memorized symbols and symbols with overmemorization from high frequency**. These contributions are novel compared to concurrent works on evaluating culture biases in LLM generations and **can be integrated with existing bias mitigation or unlearning strategies to solve culture biases**.
>
> Many works have been done on unlearning knowledge, mitigating biases, or combatting spurious patterns through influential data [1][2][3][4][5]. These works follow the pipeline of influential pattern attribution, mostly through influence function [6] or feature attribution [7], and then demote the patterns by either learning confound-invariant representations, e.g., adversarial training [8] or data-based approaches to balance the training data, e.g., counterfactual data augmentation[9]. Our discovery of symbol overmemorization can be an alternative to the first step, as it helps identify high-frequency symbols that are biasing model generations, which are the targets of demotions in the second step. Our discovery of cultures with inadequate memorized symbols can also facilitate strategies for data augmentation: from pretraining documents containing these cultures, we find symbols that are related to the culture but not generated by the models, and these will be the symbols that need augmentation.
>
> [1] Mitigating Social Biases in Language Models through Unlearning
>
> [2] Influence Tuning: Demoting Spurious Correlations via Instance Attribution and Instance-Driven Updates
>
> [3] Adversarial Unlearning: Reducing Confidence Along Adversarial Directions
>
> [4] Debiasing Machine Unlearning with Counterfactual Examples
>
> [5] Unlearn What You Want to Forget: Efficient Unlearning for LLMs
>
> [6] Understanding black-box predictions via influence functions
>
> [7] “Why should I trust you?”: Explaining the predictions of any classifier
>
> [8] Adversarial removal of demographic attributes from text data
>
> [9] Counterfactual data augmentation for mitigating gender stereotypes in languages with rich morphology
>
> >Q1. In Figure 1, the top-down order does not match the numerical order. Why are memorized symbols shown at the bottom?
>
> In Figure 1, we order the symbol categorizes based on their level of “grounding” on pretraining documents. Memorized symbols are most grounded to pretraining documents due to the fact that these symbols are generated for the culture because of high symbol-culture co-occurrence in pretraining data; generalized symbols are traced to memorized symbols as evidenced from our analysis in Section 4.4; independent symbols do not have any evidences from pretraining document that associate them with cultures, and therefore it is on the top of the pyramid.
>
> >Q2. In Figure 3, what does "overgeneralization" refer to? It is not mentioned in the text. Do you mean "overmemorization" instead? The same applies to the caption of Table 3.
>
> Thank you for catching this typo. We have changed all "overgeneralization" to "overmemorization".
>
> >Q3. How are culture-referring n-grams defined for the Document-Signal to Noise Ratio?
>
> In L220, we define culture-referring n-grams as the n-grams associated with that culture’s country name and its corresponding nationality for identifying people from that culture. For example: India and Indian; China and Chinese; South Africa and South African, etc.
>
> >Q4. Why do the memorization classification criteria differ for cases where n(C_G) > 5 and n(C_G) < 5?
>
> The memorization classification criteria differ for cases where (n(C_G) > 5) and (n(C_G) < 5) because, based on empirical observations, the Z-score becomes a misleading metric when the sample size ((n(C_G))) is small (i.e., <=5). This is because distributions with such small sample sizes often do not approximate a normal distribution well, meaning extreme Z-scores, like 2.6, are unlikely to be valid or meaningful. Therefore, we apply different criteria for small and large (n(C_G)) to account for the differences in distribution behavior and statistical reliability.

---

> > ### Author Response · Authors · 2024-11-22
> > **Response to Review (3)**
> >
> > > Q5. What do the bold texts represent in the "topic modeling keywords" column of Table 3?
> >
> > The bold texts in the "topic modeling keywords" column of Table 3 highlight key topics that are discovered in the pre-training documents, which may contribute to the phenomenon of culture overmemorization. These bolded topics serve as indicators of specific themes or issues that help explain how cultural elements, such as the concept of the hijab, might be overmemorized or transferred between cultures. For example, in the case of Iran and Saudi Arabia, the bold topics related to the hijab—such as "women," "rights," "Muslim," and "politics"—highlight the core themes in which both cultures appear in the pretraining data, where the association of “hijab” to Iran is transferred to Saudi Arabia. By identifying these key topics, we can better understand the non-symbol-related factors influencing the overmemorization process.

---

> > > ### Comment · Reviewer_UeFD · 2024-11-25
> > > **Acknowledgement of Rebuttal**
> > >
> > > Thanks for the detailed response and updated experiments. I believe authors mostly addressed my comments and I updated my assessment accordingly.

---

> > > > ### Author Response · Authors · 2024-11-25
> > > > **Thank you for your feedback!**
> > > >
> > > > Dear Reviewer UeFD,
> > > >
> > > > Thank you for your positive feedback! We are happy to hear that our rebuttal addressed your concerns.
> > > >
> > > > Sincerely,
> > > >
> > > > Paper13342 Authors

---

### Official Review · Reviewer_nfL2 · 2024-11-05

**Soundness:** 4
**Presentation:** 3
**Contribution:** 3
**Rating:** 8
**Confidence:** 3

**Summary:**

This paper describes MEMOED (MEMOrization from pretraining document), a framework designed to classify cultural symbols in LLM-generated text as either memorized or generalized.

**Strengths:**

1. Novel framework and the new problem of analyzing cultural memorization: The paper develops a systematic approach to determine if cultural symbols generated by an LLM are due to memorized data or generalization. This is a novel problem and the approach is sound and elegant.
2. Good Analysis across Cultures: The study uses data for 110 cultures on topics like food and clothing - the analysis is interesting and the conclusions are interesting as well.

**Weaknesses:**

Reliance on a Single Model: The analysis focuses solely on the OLMo-7B model and its pretraining dataset, Dolma. Its is unclear from the analysis how the conclusions would vary on other models or models of other sizes.

It is not clear to me how the definitions of what constitutes memorization (e.g. training document classification)  might change the analysis?

**Questions:**

Is it possible to do this analysis on several OLMo models?

There are some typos: and and (320)

---

> ### Author Response · Authors · 2024-11-22
> **Response to Review**
>
> Thank you for your positive review and very helpful comments! Below are our responses and we have updated the draft accordingly.
>
> > W1: Reliance on a Single Model: The analysis focuses solely on the OLMo-7B model and its pretraining dataset, Dolma. It is unclear from the analysis how the conclusions would vary on other models or models of other sizes.
>
> We quickly collected culture-conditioned generations for both food and clothing on `olmo-7b-0424` (OLMo 1.7) which is trained on Dolma 1.7. Although this is the same model family as `OLMo-7B`, Dolma 1.7 contains pretraining documents not in Dolma 1.5, and OLMo 1.7 is trained with an updated algorithm from OLMo 7B. Other models supported by the WIMBD API we used, such as `Pythia`, are not particularly capable of instruction following culture-conditioned generations, and therefore, analyzing their generations is less informative.
>
> Due to the time constraints of the rebuttal, we are unable to completely reproduce all analysis done in the paper. We analyzed 2 correlations (added to Appendix E.2):
>
> 1) Between the number of cultures independent symbol was generated for and its pre-training count:
> | Model       | Category  | Spearman Correlation | Kendall Correlation |
> |-------------|-----------|----------------------|---------------------|
> | OLMo 1.7    | Food      | 0.416                | 0.313               |
> | OLMo 1.7    | Clothing  | 0.507                | 0.362               |
> | OLMo 7B     | Food      | 0.358                | 0.260               |
> | OLMo 7B     | Clothing  | 0.521                | 0.367               |
>
> We see that even though models and training data are different, the Spearman and Kendall correlations for food and clothing remain the same (both moderate-to-strong correlations). This means that the number of cultures an independent symbol was generated for and its pretraining count is positively correlated, regardless of the model.
>
> 2) Between number of memorized symbols and pre-training counts (OLMo 1.7 is sampled on 10 cultures due to time constraint, detail see Appendix E.2):
>
> | Model       | Category  | Spearman Correlation | Kendall Correlation |
> |-------------|-----------|----------------------|---------------------|
> | OLMo 1.7    | Food      | 0.829                | 0.659               |
> | OLMo 1.7    | Clothing  | 0.591                | 0.507               |
> | OLMo 7B     | Food      | 0.670                | 0.507               |
> | OLMo 7B     | Clothing  | 0.540                | 0.421               |
>
> For this experiment, exhaustively searching for all memorized symbols of all 110 cultures requires running MEMOed on all symbol-culture pairs, which is not feasible due to the rebuttal time constraint. Nonetheless, we see that for clothing and food, the correlation of Olmo 1.7 increases despite the small number of cultures we evaluated. Therefore, the conclusion still holds that higher pretraining counts of cultures increase the number of memorized symbols in culture-conditioned generations.
>
> >W2: It is not clear to me how the definitions of what constitutes memorization (e.g. training document classification) might change the analysis?
>
> In this work, we study memorization from the perspective of cultural knowledge, and we wish to determine whether a symbol S is generated for culture C because the co-occurrence between culture C and symbol S is memorized by the model from the pretraining data. Many works on correlating model performance with knowledge memorization use the co-occurrence of input-output n-gram pairs in pretraining corpora to approximate memorization [1][2][3], and our work on using input-output relevant training documents as definition for memorization falls in this line of work.
>
> Other definitions of memorization, such as exact matches of generation [4] or influential examples discovered by influence functions [5], may only impact the categorization of which symbols are memorized. The rest of the analysis, including how memorization is correlated with pretraining frequency of cultures, as well as studies on symbol overmemorization, culture overmemorization and traceable generalization is not impacted by the definition of memorization. To examine how different definitions of memorization will impact the conclusions on memorization capabilities of models on cultural knowledge would be future work.
>
> [1] Generalization vs. Memorization: Tracing Language Models’ Capabilities Back to Pretraining Data
>
> [2] Large Language Models Struggle to Learn Long-Tail Knowledge (ICML 2023)
>
> [3] When Not to Trust Language Models: Investigating Effectiveness of Parametric and Non-Parametric Memories (ACL 2023)
>
> [4] Quantifying memorization across neural language models (ICLR 2023)
>
> [5] Studying Large Language Model Generalization with Influence Functions (Anthropic)

---

> ### Comment · Reviewer_nfL2 · 2024-11-28
> **Thanks**
>
> Thanks for the detailed response and updated experiments.

---

### Official Review · Reviewer_gSz2 · 2024-11-06

**Soundness:** 4
**Presentation:** 2
**Contribution:** 3
**Rating:** 6
**Confidence:** 3

**Summary:**

About culture-conditioned generations out of LMs, dealing with the issue that biased generations can be driven by pretraining corpus statistics.

They introduce a “symbol attribution framework” to determine if culture-conditioned symbols were memorized in training. They characterize symbols as independent, memorized, or generalized depending on whether they appear in a “culture’s generations” broadly, without a specific culture association, appear primarily in a small set, or if they appear broadly across cultures without presence in the pretraining corpora.

They use document SNR, minimum token distance, and minimum sentence distance as metrics.
- document snr is the log probability ratio of counts of culture-referent n-grams to others
- minimum token distance is the length of the shortest span of tokens between one referring to the culture and one to the symbol
- minimum sentence distance uses sentences instead of tokens in the above

They use these functions to construct a heuristic for whether a training document shows the memorizable relationship. They then characterize concepts as being overmemorized, and compare the presence of these statistics in the training data & outputs as a predictor to the agreement of human annotators that these relationships are reflective.

They claim that “traceable generalization”, ie., concepts that are not closely related according to their metrics in training data are nonetheless successfully generated, are not correlated to “memorized” concepts for a culture. There’s one or the other, for example “Mexico” contains specifically memorized queries, while Trinidad has none.

**Edit**: I have responded to the authors rebuttal, and modified my "weaknesses" section. I think the technical contributions are sound, so I have bumped my soundness up to 4 (even though I still am a bit troubled by the scope of the experiments). However, I feel that the presentation of this work is severely flawed, particularly wrt how a reader has to piece together what the experimental methodology was while reading the results. So, I am keeping my presentation score at 2 (though I contemplated dropping it to 1). I will keep my overall score at weak accept---I don't think this paper is ready but if it were to get accepted, interested researchers would be able to make their way through it. I would strongly recommend that the authors consider edits for clarity that address my complaints here in the CR if it does get accepted.

**Strengths:**

Interesting and useful topic to address.

Mildly interesting results; though I am a little unsure about the claims (see weaknesses).

Approach may generalize, not only to memorization of cultural relationships but also to memorization of other facts/information conditioned on context.

**Weaknesses:**

Presentation of relatively shallow experiments, ~limited technical novelty~, and limited scale of experiments.

~I’m not fully convinced about these definitions that are used to characterize the memorization classes; how do we know that having these statistics over some threshold means that a concept is “memorized” vs just being consistently generated?~

**EDIT:** I understand the paper better after the authors' explanations and edits. I change my mind regarding the technical novelty (which is an unfair complaint to even have in the first place even if it were accurate)

That being said, I stand by my complaint about the small scale of experiments: while many symbols are generated, and a basically comprehensive set of countries are tested, **only two prompts are used to elicit these outputs.**

Over all, my biggest complaint about the paper didn't make it in to my review, but it's the **poor presentation of the method**. I believe it is a serious problem that key details of the experiments such as "how many prompts? how many cultures? how were the symbols extracted from the outputs?" are not clearly lain out before the results. Additionally, the presentation of the methods suffers from a lot of superfluous mathematical notation that clouds clarity, with symbols that once again aren't introduced until *after* an equation is read, requiring considerable backtracking.

See my response to the reviewers.

**Questions:**

This review is a little low confidence; I'm open to changing my mind.

Please clarify any misunderstandings I have, and elaborate on my concern about the definitions?

Why were the classes of concept chosen?

---

> ### Author Response · Authors · 2024-11-22
> **Response to Review (1)**
>
> Thank you for your positive review and very helpful comments! Below are our responses and we have updated the draft accordingly.
>
> > W1: Presentation of relatively shallow experiments, limited technical novelty, and limited scale of experiments.
>
> We politely disagree.
>
> We provided **in-depth experiments** that explored four reasons why a symbol is generated by an LLM in culture-conditioned generations, finding significant correlations of model behavior with pretraining data, such as:
>
> 1) positive correlation between **the number of memorized symbols for a culture** and the **count of documents in which the culture appears in the pretraining corpora**, suggesting that it is more difficult for models to memorize symbols for cultures appearing in less frequency in pretraining data. This finding *urges future work to augment the pretraining data for long-tail cultures* or *improve the pretraining algorithm to improve memorization in low-frequency settings*.
>
> 2) positive correlation between the **average ratio of an independent symbol to all memorized symbols** and the **number of cultures that the independent symbol is generated for**, suggesting that the reason why a symbol that is not associated with any culture can nonetheless be generated for many cultures is that the symbol itself has a very high frequency in pretraining data. This finding urges further research to *increase model compliance to prompts against high-frequency but less desirable generations*.
>
> 3) positive correlation between **how often a culture’s memorized symbol is generated for
> some other cultures** and **the number of appearances in topic-related pre-training documents**, suggesting that the model is more likely to generate memorized symbols of cultures with higher frequency, despite the generation not being conditioned on those cultures. This finding suggests that *models are able to generalize to new relations from pretraining data given related concepts*.
>
> Our paper’s **technical novelty** includes 1) **developing MEMOed**, a new systematic approach for determining whether symbols generated by an LLM are due to memorized co-occurrences in pretraining data, overmemorization from high-frequency entities or cultures, and generalization that can be traced to concepts that are related to the symbol. As the reviewer also mentioned, our approach can be **generalized to other memorization scenarios** where facts/information are generated conditioned on context. 2) **introducing Document-SNR** combined with token and sentence distance as a novel metric for determining document relevance to a pair of concepts, which, by our knowledge, **is the first work to introduce an SNR-like metric to pretraining data analysis**.
>
> Our paper’s **scale of experiments** is **by no means small**: we studied **110 cultures**, where model generations contain **2370** unique symbols for food and **1002** for clothing. In total, we have found **19.4 million documents** to be relevant to the memorization of symbol-culture pairs in food and **7.85 million documents** to be relevant to the memorization of symbol-culture pairs in clothing.

---

> > ### Author Response · Authors · 2024-11-22
> > **Response to Review (2)**
> >
> > > W2: I’m not fully convinced about these definitions that are used to characterize the memorization classes; how do we know that having these statistics over some threshold means that a concept is “memorized” vs. just being consistently generated?
> >
> > 1) We use training document relevancy, contributory score, and z-score threshold to **collectively define memorization**. We define the training document classification (L254) metric to classify whether a pretraining document contributes to the memorization of the relationship between a symbol S and culture C. Then, the contributory score calculates the number of documents that contain symbol S that are contributory to the memorization of the relationship between symbol S and culture C. We then form a distribution of the contributory score of all cultures for which the symbol S is generated. The intuition behind this is that if the distribution of the contributory scores of all cultures is flat, then the symbol is not distinguishably associated with any of the cultures; only if the distribution of the contributory scores is spiked at a few cultures, then the symbol is associated with those cultures. Therefore, we find cultures that are “outliers” within the distribution using z-score=2.6 as a threshold as z-score=2.6 means that the value is above 99.5 percentile [1], effectively suggesting that the culture is distinguishable from other cultures with its association with the symbol.
> >
> > *To answer your question on the selection of z-score threshold*, in Appendix E.3 we added a study on **whether selecting a different z-score threshold would change the conclusions of MEMOed** on memorized symbols for all cultures. We perform an ablation study on setting the z-score to 2, which statistically means that the value is about 97.7 percentile. Empirically, a z-score below 2 does not indicate outliers, so we focus our ablation analysis only on cases where the z-score is 2.
> >
> > When z=2, we get still get a moderate-to-strong correlation between 1) the number of memorized symbols for a culture and 2) the count of documents in which the culture appears in the pretraining corpora: for clothing, we obtain a spearman correlation of 0.569 and a Kendall correlation of 0.445; food food, we obtain a spearman correlation of 0.688 and a Kendall correlation of 0.519. This correlation is lower but similar to the original correlations found for z=2.6 (food: Spearman=0.670 and Kendall=0.507; clothing: Spearman=0.540 and Kendall=0.421), showing that **our conclusion on the relationship between a culture's memorized symbols and the culture's frequency in pretraining data is robust to different z-score threshold**.
> >
> > In addition, we examine how lowering the z-score from 2.6 to 2 changes memorized symbols discovered for each culture. We compare each metric's agreement with human evaluation on clothing: when z=2.6, the weighted F1 score is 0.845, and when z=2, the weighted F1 score is 0.840. We can see that z = 2 has a slightly lower agreement with human categorization, suggesting that **additional symbols that are marked as memorized symbols when z=2 are non-emblematic symbols according to human culture experts**.
> >
> > 2) Our metric **does not count the number of times that a symbol is repeatedly generated for a given culture**; rather, as long as a symbol is generated for a culture, however many times during sampling, we classify memorization or not using the symbol’s relevant co-occurring pretraining document count with the culture in the pretraining data. In addition, a symbol "being consistently generated" may not suggest memorization: we find that the **most "consistently generated" symbols are the results of symbol overmemorization**. We took the average number of repetitions of the top 10 most consistently generated independent symbols and the average number of repetitions of all memorized symbols for each culture and averaged this measure across cultures. Results show that for clothing, the average repetition of symbol overmemorization is 50.59 times, and for food, the average repetition of symbol overmemorization is 52.10 times. In contrast, the average repetition of memorized symbols is 2.46 and 15.75 for clothing and food, respectively. So, we can see that independent symbols are substantially more consistently generated than the maximally repeating memorized symbols.
> >
> > [1] https://www.sjsu.edu/faculty/gerstman/EpiInfo/z-table.htm
> >
> > > Q1: Why were the classes of concept chosen?
> >
> > What the reviewer meant by "classes of concept" is a little unclear, but right now, we interpret the question as "Why was this study focused on the topics of food and clothing?" If we should have interpreted otherwise, please definitely let us know.
> >
> > As mentioned in L194, we choose food and clothing among all topics introduced in Culture-Gen because the food and clothing of each culture vary significantly, and the knowledge of food and clothing entities must be obtained through memorization from pretraining data.

---

> ### Author Response · Authors · 2024-11-25
> **Friendly reminder to respond to author rebuttal**
>
> Dear Reviewer gSz2,
>
> Thank you again for your review! We are happy to hear that you appreciated the usefulness of our research problem and generalization of our approach.
>
> Based on your thoughtful feedback, we wrote a detailed rebuttal covering the following points:
>
> - Justification on our technical contribution, scale of experiments and depth of analysis
> - Clarification on our definition of memorization and how it is different from measuring generation consistency
>
> We would love to hear your thoughts about our rebuttal, including whether it sufficiently addresses your concerns and questions. If you believe that our rebuttal is satisfactory, it would be great if you could consider increasing your score. Any feedback is welcome and greatly appreciated!
>
> Sincerely,
>
> Paper13342 Authors

---

> > ### Author Response · Authors · 2024-11-26
> >
> > Dear Reviewer gSz2,
> >
> > Just wanted to follow up on our previous message! As you know, we are quickly approaching the draft update deadline Nov 27 EOD.
> >
> > In light of this, we would love to hear your thoughts about our rebuttal, including whether it sufficiently addresses your concerns and questions, and whether you would like to see further edits made.
> >
> > If you believe that our follow-up rebuttal is satisfactory, it would be great if you could consider increasing your score. Any feedback is welcome and greatly appreciated!
> >
> > Sincerely,
> >
> > Paper13342 Authors

---

> ### Comment · Reviewer_gSz2 · 2024-12-03
> **Rebuttal acknowledgement**
>
> Thank you for the detailed response! Putting "limited novelty" in the review was a lazy and out-of-line comment on my part, so I would understand even an impolite disagreement!
>
> "Shallow experiments" was really meant to convey my dissatisfaction with the *level of detail provided for the experiments in the main text*. Instead, a lot of space is wasted on kind of superfluous math, which is a common trap for ICLR/NeurIPS papers imo.
>
> 1.5 pages are devoted to an overly complicated explanation of basically "we use thresholds of our three scores to select documents that count toward memorization classification, and then we use the ratio of counted documents for the [symbol, culture] over all documents containing the symbol to determine if the symbol is memorized for a culture" using lots of unskimmable equations with unclearly defined (mathematical) symbols (for ex, $r(D,Q)$ is only defined as "considered relevant... i.e.$r(D,Q)=1$". You never *say* what $r$ even is!
>
> Despite those 1.5 pages, not a single line in the entire main text clearly states the process by which the generated culture/symbol pairs are elicited or annotated! (though I appreciate you adding this to the appendices) Generally, it is good to tell the readers **what the experiments are, clearly** *before* you start giving them results. Instead I have to go back and forth, find the number of cultures tested then go back up, find the number of symbols tested, number of generations, etc. **You need to open the results with a clear explanation of what the experiments are; something like "using the following prompts we generated X  total sentences for Y concepts in Z cultures and then [annotation protocol]." Even now after going through this paper a few times, I just now understand how even the symbols are produced: **you use the final tokens outputted from two template sentences** (uncovered via detective work and the new prompts appendix section). In the main text, this should be very clearly highlighted!
>
> **Overall, I am reluctant to recommend this paper for publication due to these presentation issues.**
>
> **That being said, I would like to acknowledge the technical novelty. (but only using it on two prompt templates for the main cultural symbol generation process is a little to limited imo) I think this is an exciting approach and hope to see it published, but *please* work on writing a more accessible paper that doesn't require so much backtracking to understand, and clearly explain the methodology for your empirical results in the main text.**
>
> **I have raised my soundness score, but will keep the others. I have updated my review text.**
>
> (Sorry to acknowledge the rebuttal so late!)

---

> ### Comment · Reviewer_gSz2 · 2024-12-03
> **On superfluous math**
>
> To expand on my complaint about the overuse of mathematical notation, here's an example: (I excerpted this out from my first rebuttal response to make it easier to follow)
>
> Introducing equations for the metrics is ok, but going so far as $r$ for relevance (which only ever takes a *binary value*) using the formulas in l255-257, and $m$ in l296-299 is just extremely reader unfriendly.  For me reading this was basically:
> > ok, $Q$ is a pair of $[C,S]$, ok so $r$ is .... ok $r$ means relevant so $r(D,Q)$ is what? A culture or symbol? Oh right $Q$ is $[C,S]$ so $r(D,Q)$ means that document $D$ is $r$elevant to culture $C$ and $S$ in $Q$...
>
> You could easily (and more clearly!) convey document relevance by just saying something like
> > We consider a training document to be *relevant* in determining if a symbol $S$ is memorized for culture $C$ if the either the token or sentence distance are sufficiently low along with document SNR requirements, or [l260-265]
>
> **Using this weird notation of setting a binary value to variable $r$ that never gets reused anywhere to 1 instead of just saying "under this condition a document is relevant" is very reader unfriendly and is emblematic of presentation issues throughout the paper.**
>
> This writing style basically makes p5 and 6 (explaining the method) pretty burdensome to follow, and frankly feels forced to impress the modal ICLR reviewer.

---

> > ### Author Response · Authors · 2024-12-03
> > **Thank you for response to rebuttal**
> >
> > Dear reviewer gSz2,
> >
> > Thank you very much for the updated review and detailed feedback. We really appreciate your suggestions on a clearer presentation of the methodology and experiments, and will incorporate the following changes that you suggested in our camera-ready version if accepted:
> >
> > 1) shrink the mathematical notations in 3.3, on "criterion for training document classification" and "criterion for memorization classification", to concise language with minimal variables.
> >
> > 2) move the experiment setup from the Appendix to the main text, highlighting the number of cultures, prompt templates, number of generations, as well as how culture symbols are extracted from generations.
> >
> > Please let us know if you have any further suggestions on how we could improve the readability of our paper. We greatly value your feedback.
> >
> > Sincerely,
> >
> > Paper13342 Authors

---

### Official Review · Reviewer_PsWQ · 2024-11-09

**Soundness:** 2
**Presentation:** 3
**Contribution:** 2
**Rating:** 5
**Confidence:** 3

**Summary:**

This paper introduces a framework, called MEMOED, to determine whether culture related entities, refered to as symbols, are resulted from memorization or generalization of LLMs based on the pretraining data. This study defines three categories of symbols, i) independent symbols, ii) culture-specific symbols, referred to as memorized symbols, and iii) symbols generalized across certain cultures, referred to as generalized symbols. Various experiments are conducted using OLMO on 110 cultures to understand how OLMO's performance is affected by memorization and generalization.

**Strengths:**

* This paper selects a good research problem, which is to understand how the frequencies of certain culture related concepts or entities in the pre-training data influence the model performance, in particular from the perspective of generalization and memorization.
* The high-level ideas to discuss about independencies, memorization and generalization are reasonable.
* The dataset covers over 100 cultures.

**Weaknesses:**

* The definitions of the following concepts and their justification are unclear unclear to me.
    * What is a symbol? Do they cover all linguistic variations of the same entity or concept?
    * How culture is defined? Why it is represented as a combination of country and natonality. The literature in social science has already defined culture. There could be more than one cultures in a country. Would a representation of country and nationality be overly simplied?
    * How to justifiy the definition of memorization through Equation (1)? Why it makes sense?
    * How generalization is defined and why?
* It lacks of justification of the formula for r(D, Q) in Page 5, as well as the measure for memorization. Why log ratio is preferred over the standard techniques, e.g. statistical dependencies? There are often various ways to convey an entity or a concept. How are linguistic variations captured with this measure? As this measure is used together with the contribution score and z-score to determine if a symbol is memorized. There is no empirical evidence or theoretical justification showing that this measure indeed meets the expectation.

**Questions:**

* There could be an alternative way to convey symbol and culture overmemorization, if the purpose is to show that certain entities occur more often in model outputs that those observed in the pre-training data.
* How do you ensure the quality of annotations using culture experts?
* How symbols are collected? Is there a systematic way to sample data from the 110 cultures?

---

> ### Author Response · Authors · 2024-11-22
> **Response to Review (1)**
>
> Thank you for your helpful comments and insightful questions! *We have grouped your feedback and questions into some common themes. Please let us know if any of your questions remain unaddressed.* Based on your feedback, we have updated the draft and provided the responses below.
>
> >W1: Justification for Equation (1).
>
> First, we would like to clarify that Equation (1) is not our metric to determine memorization; rather, this equation defines one metric that we use to classify the relevance of one pretraining document D to a culture C. We name this metric “Document-Signal to Noise Ratio” because it is inspired by the Signal-to-noise ratio (SNR or S/N) measure that is used in science and engineering that compares the level of a desired signal (meaningful input) to the level of background noise (meaningless or undesired input) [1]. SNR is defined as the ratio of signal power to noise power, and a ratio higher than 1:1 indicates more signal than noise.
>
> In our work, we categorize the desired signal (meaningful input) as the count of culture C in the pretraining document and background noise (undesired input) as the sum of counts of all other cultures (plus a small epsilon). We compute base-2 log over this ratio mainly for better interpretability as a d_SNR >= 0 means the count of culture C is at least as many as the sum of counts of all other cultures in this document, suggesting that this document is highly relevant to the culture.
>
> [1] https://en.wikipedia.org/wiki/Signal-to-noise_ratio
>
> >W2: Justification for r(D,Q) in Criterion for Training Document Classification
>
> r(D, Q) = 1 means that our MEMOed method classifies a pretraining document D as relevant to both symbol S and culture C, given that symbol S appears in the generation that is conditioned on culture C. There are two cases where we r(D, Q) = 1.
>
> In the first case, d_SNR >= 0 means the count of culture C is at least as many as the sum of counts of all other cultures in this document, suggesting that this document is highly relevant to the culture, and d_TOK <= 2048 means the subtoken distance between culture C and symbol S is shorter than the context length of the language model (here we are using olmo-7b, but this can be adjusted for other models accordingly), and therefore enables the culture C and symbol S to appear in the same example during pretraining and ensure that their co-occurrence gets learned by the model.
>
> In the second case, d_SENT <=2 means that the culture C and symbol S are occurring in adjacent sentences, suggesting that the local context is highly likely to discuss the relationship between culture C and symbol S and that this context is helpful for the model's learning of the relationship.  We added an additional condition of d_SNR between -1 and 0 so that the global context (pretraining document) is also somewhat about culture C and not entirely irrelevant.
>
> >W3: Justification of measure for memorization.
>
> In this work, we study memorization from the perspective of cultural knowledge, and we wish to determine whether a symbol S is generated for culture C because the co-occurrence between culture C and symbol S is memorized by the model from the pretraining data. Many works on correlating model performance with knowledge memorization use the co-occurrence of input-output n-gram pairs in pretraining corpora to approximate memorization [1][2][3].
>
> We build on top of this co-occurrence perspective to define memorization. Contributory Score calculates the number of documents that contain the symbol S and that contribute to the memorization of the relationship between the symbol S and culture C. We then form a distribution of the contributory score of all cultures for which the symbol S is generated. The intuition behind this is that if the distribution of the contributory scores of all cultures is flat, then the symbol is not distinguishably associated with any of the cultures; only if the distribution of the contributory scores is spiked at a few cultures, then the symbol is associated with those cultures. Therefore, we find cultures that are “outliers” within the distribution using z-score=2.6 as a threshold as z-score=2.6 means that the value is above 99.5 percentile [4], effectively suggesting that the culture is distinguishable from other cultures with its association with the symbol.
>
> [1] Generalization vs. Memorization: Tracing Language Models’ Capabilities Back to Pretraining Data
>
> [2] Large Language Models Struggle to Learn Long-Tail Knowledge (ICML 2023)
>
> [3] When Not to Trust Language Models: Investigating Effectiveness of Parametric and Non-Parametric Memories (ACL 2023)
>
> [4] https://www.sjsu.edu/faculty/gerstman/EpiInfo/z-table.htm

---

> > ### Author Response · Authors · 2024-11-22
> > **Response to Review (2)**
> >
> > >Q1: What is a symbol? Do they cover all linguistic variations of the same entity or concept?
> >
> > Our definition of a symbol is an n-gram that represents some entity or concept related to the topic of interest, e.g., food or clothing. Linguistic variations of the same entity will be considered as different symbols (such as singular or plural forms)
> >
> > >Q2: How culture is defined? Why is it represented as a combination of country and nationality. The literature in social science has already defined culture. There could be more than one cultures in a country. Would a representation of country and nationality be overly simplified?
> >
> > We recognize that diverse cultures exist within one country (or region), but we set the granularity of cultures at the level of countries or regions as the analysis of cultures mainly adopts this granularity, both in culture bias studies in large language models [1][2], and social science works such as World Value Survey[3]  and Hofstede Cultural Dimensions[4].
> >
> > For counting n-grams that represent the culture, we consider references to both the country and nationality so as to encompass all pretraining documents that mention the culture. However, our approach can be applied at a more fine-grained level, such as for different cultures within a country, because the data collection process and memorization classification process stay the same.
> >
> > [1] CultureLLM: Incorporating Cultural Differences into Large Language Models
> >
> > [2] Towards Measuring and Modeling “Culture” in LLMs: A Survey
> >
> > [3] The world values survey. The Wiley-Blackwell Encyclopedia of Globalization
> >
> > [4] Culture's Consequences: International Differences in Work-Related Values
> >
> > >Q3: There are often various ways to convey an entity or a concept. How are linguistic variations captured with this measure?
> >
> > In our paper, we are only focusing on the surface forms (reference) to the entities or concepts, not the entities behind them. Memorization of an entity while considering all its variations in names requires a different dimension of analysis, and we do not consider this complication in our paper.
> >
> > >Q4: How generalization is defined and why?
> >
> > We define generalization as when a symbol S is generated for culture C but there are not enough relevant pretraining documents that contribute to the memorization of their relationship. In our paper, we study traceable generalization, where even though there are not enough relevant pretraining documents contributing to memorization, the generated symbol can be traced to the definition of another symbol that is memorized from pretraining data.
> >
> > >Q5: There could be an alternative way to convey symbol and culture overmemorization if the purpose is to show that certain entities occur more often in model outputs than those observed in the pre-training data.
> >
> > We would like to first clarify our goal of studying symbol overmemorization and culture overmemorization, as defined in Section 3.4: we do not claim to compare the occurrence of certain entities in model outputs with the occurrence of these entities in pretraining data; rather, we aim to explain why certain entities are generated for a culture even though there are not enough pretraining documents containing co-occurrence of these entities with these cultures.
> >
> > Symbol overmemorization results from the model being biased toward symbols with high frequency in pretraining corpora, and culture overmemorization results from the model being biased toward cultures with high frequency in pretraining corpora. Therefore, the model bias makes retrieving these symbols easier during generations than retrieving memorized symbols of cultures with lower frequency.
> >
> > Our experiment results in Section 4.3 support this claim. For symbol overmemorization, we have found positive correlations between the number of cultures an independent symbol is generated for with 1) the number of pretraining documents an independent symbol appears in and 2) the average ratio of the pretraining frequency of an independent symbol to all memorized symbols.
> >
> > For culture overmemorization, we conducted an additional experiment (Appendix G.1) to show that cultures whose memorized symbols are generated for other cultures are concentrated to those with high frequency in pretraining documents. 34 cultures in clothing and 86 cultures in food have at least one of their memorized symbols generated for other cultures. Upon calculating correlation of the frequency of culture overmemorization for each culture and the culture’s frequency in pretraining documents, we observe a high correlations for both clothing (Spearman ρ = 0.763, Kendall τ = 0.574) and food (Spearman ρ = 0.716, Kendall τ = 0.531). These results suggest that cultures frequently overmemorized are also those more commonly appearing in pre-training documents. We show this correlation through scatter plots for both clothing and food in Figure 14.

---

> > > ### Author Response · Authors · 2024-11-22
> > > **Response to Review (3)**
> > >
> > > >Q6: How do you ensure the quality of annotations using culture experts?
> > >
> > > As mentioned in Section 4.2, we carefully selected culture experts from Prolific, the crowdsourcing annotation platform with the highest quality of annotators. To ensure the reliability of the annotation, we selected 8 cultures (American, Chinese, Filipino, Indian, Ghanaian, Japanese, Mexican, and Vietnamese) that have more than 25 active annotators who were born in the culture but are currently in the US (this is for payment requirement). We provide detailed explanations of each option as well as example annotations. Each example is annotated by 3 annotators, and we use the majority vote as the label. Details of the questionnaire are in Appendix D. We also include one attention check question for every questionnaire to ensure the culture experts are paying attention to the questions.
> > >
> > > >Q7: How are symbols collected? Is there a systematic way to sample data from the 110 cultures?
> > >
> > > Symbols are collected following the approach of Culture-Gen[1] (L192), but on `OLMo-7b`. We added details of the symbol collection process in Appendix B.1.
> > >
> > > We prompt the model in a continuing generation task where we use the following topic-wise prompts:
> > >
> > > - Food: My neighbor is [culture]. At dinner, [he/she/my neighbor] probably likes to eat
> > > - Clothing: My neighbor is [culture]. [he/she/my neighbor] is probably wearing
> > >
> > > We sample 100 generations for male, female, and gender-agnostic settings, and thus, for each culture, we get 300 generations.
> > > Language models usually complete this prompt with one or more symbols. We took this completion and used `LLAMA-3-70b-instruct` to extract the symbols from this generation and cache them locally. The prompt for extracting symbols can be found in Culture-Gen.
> > >
> > > [1] CULTURE-GEN: Revealing Global Cultural Perception in Language Models through Natural Language Prompting

---

> ### Author Response · Authors · 2024-11-25
> **Friendly reminder to respond to author rebuttal**
>
> Dear Reviewer PsWQ,
>
> Thank you again for your review! We are happy to hear that you appreciated the importance of our research problem and the depth of our data.
>
> Based on your thoughtful feedback, we wrote a detailed rebuttal covering the following points:
>
> - Justification of Equation 1 and definition of memorization and generalization
> - Clarification of the purpose of studying symbol and culture overmemorization
> - Clarification of definitions of symbols and cultures
>
> We would love to hear your thoughts about our rebuttal, including whether it sufficiently addresses your concerns and questions. If you believe that our rebuttal is satisfactory, it would be great if you could consider increasing your score. Any feedback is welcome and greatly appreciated!
>
> Sincerely,
>
> Paper13342 Authors

---

> > ### Author Response · Authors · 2024-11-26
> > **Another friendly reminder to respond to author rebuttal**
> >
> > Dear Reviewer PsWQ,
> >
> > Just wanted to follow up on our previous message! As you know, we are quickly approaching the draft update deadline Nov 27 EOD.
> >
> > In light of this, we would love to hear your thoughts about our rebuttal, including whether it sufficiently addresses your concerns and questions, and whether you would like to see further edits made.
> >
> > If you believe that our follow-up rebuttal is satisfactory, it would be great if you could consider increasing your score. Any feedback is welcome and greatly appreciated!
> >
> > Sincerely,
> >
> > Paper13342 Authors

---

### Author Response · Authors · 2024-11-22
**General Response**

We thank all of the reviewers for their thoughtful feedback and recognition of our paper’s contributions! We are delighted to see that the reviewers appreciated our work’s **novelty** (nFL2, UeFD), **impact**(PswQ, gSz2, UeFD), **scope of study** (PswQ, D6x9), **generalizability** of our approach to other problems (gSz2, D6x9), and **depth of analysis** (nFL2, UeFD).

In response to the reviewers’ comments/questions, we have addressed the following items in our rebuttal and updated paper (changes marked in orange):

- Demonstrated the **generalizability of our approach, analysis and conclusion** on another `OLMo` model `OLMo-7B-0424` and its pretraining data Dolma 1.7. The result is in Appendix E.2 (nFL2, UeFD, D6x9)
- Showed that our analysis on memorization for under-represented cultures is **robust** to z-score threshold by conducting ablation studies comparing z=2 and z=2.6. The result is in Appendix E.3 (gSz2)
- Discussed how our work’s findings can be **integrated into existing unlearning/bias mitigation approaches** to solve biases in culture-conditioned generations (UeFD, D6x9)
- Added correlation statistics between culture overmemorization phenomena and the pretraining frequency of cultures. The result is in Section 4.3 and Appendix G (PswQ)
- **Summarized our work’s novelty and contribution (below)**. We will update the paper to better convey this (gSz2)
- Elaborated on the design choices of our measurement of memorization, training document classification and definition of generalization (PswQ, gSz2, UeFD)
- Elaborated on the conclusions from our studies on symbols overmemorization and culture overmemorization (PswQ)
- Discussed quality control of culture expert annotations (PswQ)
- Discussed definitions of symbols and cultures and collection of culture-conditioned generations (PswQ)

**Summary of Contribution**

1. We propose **MEMOed**, a novel pretraining data attribution framework that grounds memorization classification on relevant pretraining documents

- **MEMOed’s training document classification module** uses a novel metric, *Document-Signal-to-Noise-Ratio (d_SNR)*, to determine whether a pretraining document contributes to memorization of symbol-culture correlation. MEMOed takes into account distance between symbol and culture n-grams and the density of information in the pretraining document, adhering to realistic pretraining situations.
- **MEMOed’s memorization classification module** is optimized to find the *most correlated culture to a given symbol*. To ensure that a culture is distinguishably associated with the symbol apart from other cultures, we propose contribution score, the proportion of pretraining documents relevant to the symbol-culture pair over all symbol occurrences, and innovatively employ z-score threshold to find the culture with significant association with the symbol.

2. We propose a **nuanced categorization of memorization and generalization** phenomena, find **significant positive correlations** between each phenomenon and pretraining data frequency

- We demonstrate positive correlation between the *number of memorized symbols* for a culture and the *count of documents* in which the culture appears in the pretraining corpora
- We demonstrate positive correlation between the average *ratio of an independent symbol pretraining occurrence to all memorized symbols* and the number of cultures that the independent symbol is generated for
- We demonstrate positive correlation between *how often a culture’s memorized symbol is generated for some other cultures* and the *number of appearances in topic-related pre-training documents*
- We demonstrate that *a small set of independent symbols* unassociated with any culture comprise a *significant proportion of the total responses*, demonstrating the need to mitigate model’s overconfidence in high frequency symbols.

---

### Meta-Review · Area_Chair_JvFJ · 2024-12-16

**Metareview:**

This paper introduces MEMOED, a novel framework for analyzing how pretraining data contributes to cultural biases in large language models (LLMs). By categorizing symbols as either memorized or generalized, the study reveals how pretraining data biases LLM outputs. Specifically, models tend to overmemorize symbols from overrepresented cultures while struggling with culture-specific symbols from underrepresented ones. Based on experiments conducted with the OLMo-7B model across 110 cultures, the work highlights significant limitations in current LLMs and underscores the need for improved methods to enhance cultural diversity and accuracy in LLM generations.

Strengths:

* The paper addresses a highly relevant and underexplored problem by systematically analyzing cultural memorization and generalization in LLMs.
* The analysis is thorough, encompassing data from 110 cultures across diverse topics such as food and clothing.
* The study's broad scope and compelling findings on overmemorization have significant implications, not only for addressing cultural biases but also for advancing our understanding of LLMs. These insights could provide a strong foundation for future research.
* The paper demonstrates technical novelty with MEMOED, an approach to determine whether symbols generated by an LLM are derived from memorized co-occurrences in pretraining data, overmemorization of high-frequency entities, or generalization. This method could have applications in other contexts.

Weaknesses:

* Several reviewers initially raised concerns about the limited scope of the experiments, particularly the use of only one model. These concerns were largely addressed during the discussion period, with additional experiments conducted using other models.
* Clarity issues persist, and the presentation of the method needs significant improvement.

After discussion, the reviewers reached a general consensus to accept the paper, and I support this recommendation. I would highly recommend the authors to address the remaining concerns about clarity for the camera-ready paper.

**Additional Comments On Reviewer Discussion:**

The discussion primarily focused on two aspects: (1) the thoroughness of the experiments and (2) issues with clarity and presentation.
While concerns regarding the thoroughness of the experiments were mostly addressed, it appears that the presentation could—and should—be significantly improved.

Reviewer PsWQ was unable to participate in the discussion, but the other reviewers and I are satisfied with the authors' response to Reviewer PsWQ's feedback.

---

### Decision · Program_Chairs · 2025-01-22

Accept (Poster)